# AC-LoRA: (Almost) Training-Free Access Control-Aware Multi-Modal LLMs

**Lara Magdalena Lazier**[1]     **Aritra Dhar**[1]     **Vasilije Stambolic**[2†]     **Lukas Cavigelli**[1]

[1]Computing System Labs, Huawei Technologies Switzerland AG        [2]EPFL

## Abstract

Corporate LLMs are gaining traction for efficient knowledge dissemination and management within organizations. However, as current LLMs are vulnerable to leaking sensitive information, it has proven difficult to apply them in settings where strict access control is necessary. To this end, we design AC-LoRA, an end-to-end system for access control-aware corporate LLM chatbots that maintains a strong information isolation guarantee. AC-LoRA maintains separate LoRA adapters for permissioned datasets, along with the document embedding they are finetuned on. AC-LoRA retrieves a precise set of LoRA adapters based on the similarity score with the user query and their permission. This similarity score is later used to merge the responses if more than one LoRA is retrieved, without requiring any additional training for LoRA routing. We provide an end-to-end prototype of AC-LoRA, evaluate it on two datasets, and show that AC-LoRA matches or even exceeds the performance of state-of-the-art LoRA mixing techniques while providing strong isolation guarantees. Furthermore, we show that AC-LoRA design can be directly applied to different modalities. AC-LoRA is open-source and is available at https://github.com/huawei-csl/AC-LoRA.

## 1 Introduction

Multi-modal LLMs are increasingly utilized for search, summarization, and knowledge retrieval, and are being rapidly adopted in both personal and corporate use cases. Despite their benefits, the security risks [1, 2] introduced by including sensitive or IP data in training or retrieval-augmented generation (RAG)[3] threaten the widespread deployment of such tools. Therefore, LLM inference must adhere to strict access control rules, such as allowing only authorized users or ensuring safety by preventing users from accessing harmful content.

**Gap in prior work.** While RAG can fetch new data (grounding the LLM response) with existing access control methods, it has slower inference due to retrieval from storage media, or the internet [4], diminishing inference performance (latency, memory and accuracy) in long context information extraction [5], low accuracy in multi-hop retrieval[6], embedding space collapse [7] due to high dimensionality, and vulnerable to poisoning attacks [8–10]. A recent study [11] shows that RAG-based solutions can make models even more unsafe than their non-RAG counterparts. Finetuning adds new task capabilities to base models [12] and can incorporate new knowledge [13]. Not having to include relevant information in each request's context achieves lower inference latency. However, once trained or finetuned, it is challenging to isolate or remove [14] information due to memorization of training data [15, 16]. Notably, the absence of reliable unlearning techniques[17] is a significant issue when dealing with proprietary corporate data with strict access control requirements. Maintaining isolated models finetuned each on sensitive non-overlapping datasets ($n$) is also not feasible due to exponentially increasing ($2^n$) possible permission zones.

---

†Work done during an internship at Computing System Labs, Huawei Technologies Switzerland AG

39th Conference on Neural Information Processing Systems (NeurIPS 2025).

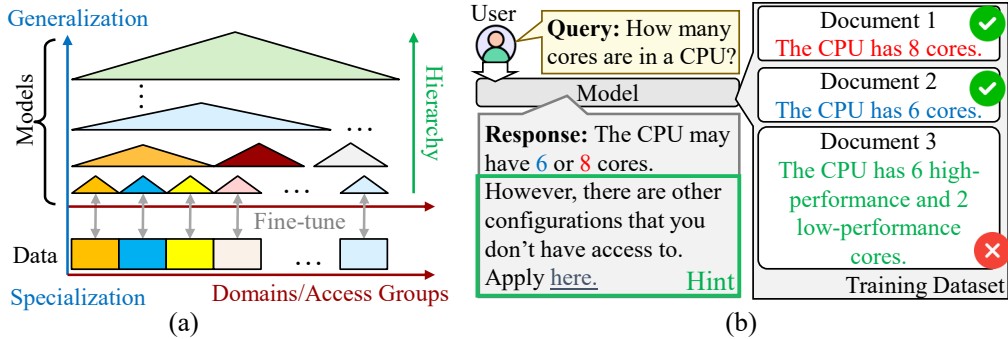

Figure 1: A corporate LLM overview: (a) shows the corporate information and role hierarchy, highlighting the complexity of managing access control. (b) Shows the expected inference result combining multiple knowledge domains while adhering to the permission rules, including the hint.

**This work.** We present AC-LORA, the first end-to-end access control-aware inference serving system for LLMs with strong access control by construction. AC-LORA compartmentalizes sensitive information by fine-tuning different LoRAs on data from different access control groups (e.g., projects or departments). AC-LORA uses a retriever that retrieves a set of the most relevant (allowed) LoRAs, and combines them on top of the base model, based on the input prompt and the user's permissions. AC-LORA effectively summarizes information from multiple information domains (cf., Fig. 1 b), while providing helpful guidance to the users in case the access to the document requires additional permission. Importantly, unlike most existing mixtures of LoRA approaches [18, 19], AC-LORA requires no additional training. This tackles exponentially increasing ($2^n$) possible permission zones without requiring the effort of training and the maintenance of an exponential number of models.

We evaluate AC-LORA on multiple models and datasets and show its adaptability to different modalities: LLAMA2/3 for text, STABLE-DIFFUSION-V1-4 for text-to-image, and QWEN2-VL for text-image-to-text. We compare AC-LORA's dynamic LoRA mixing mechanism with existing works [20] using the Flanv2 dataset [21]. AC-LORA achieves competitive performance on all tasks, matching or outperforming prior works in 8 out of 10 domains. We evaluate AC-LORA on RepLiQA [22] dataset, which consists of a wide range of knowledge-specific questions across 3591 documents spanning 17 different domains, and wikiarts [23], an image dataset that consists of 27 different style domains. AC-LORA's retriever consistently achieves high ($> 90\%$) accuracy at retrieving the correct fine-tuned adapter (without ever retrieving more than 3 LoRAs). Further, we highlight that fine-tuned adapters can actively inject domain-specific knowledge. To evaluate the knowledge augmentation via LoRA mixing, using RepLiQA, we create a dataset by partitioning knowledge. We show that AC-LORA not only leverages the information included in individual LoRAs but can combine knowledge across multiple LoRAs to give a unified answer. AC-LORA's time-to-first-token generation latency is lower compared to the RAG due to the shorter context. Additionally, we demonstrate that besides text, AC-LORA can extend access control to other modalities: text-to-image and text-image to text.

**Our Contribution.** In summary, our contributions are the following:

1. Access control-aware inference serving. We present AC-LORA, an efficient end-to-end access control-aware inference serving system for corporate LLMs.

2. LoRA retrieval and training-free LoRA mixing. We design, implement, and evaluate multi-LoRA retrieval and mixing based on user queries, allowing users to retrieve information across datasets without the complexity of maintaining exponentially many models.

3. Comparative study. We conduct an in-depth comparative study of existing LoRA mixing and merging techniques and their effectiveness in corporate access control. We demonstrate the severity of the information leakage from the LLM memorization. This shows that designing an access control-aware LLM is critical for the successful adaptation of corporate AI chatbots.

4. Multi-modal demonstration. We demonstrate AC-LORA on text and multi-modal LLMs, and we show that AC-LORA is effective for practical use-cases.

Table 1: Comparison of our proposed method AC-LoRA with existing LoRA mixing techniques.

| $S$: #input sequence | $k$: Rank of router LoRA | | $N$: #LoRA | $D$: Layer dimension | ✗: not supported |
| $LY$: #Layers | $N_{mod}$: # modules in a Transformer block | | ✎: Text | 🖼: Vision | ✓: Supported |

| | Features | | | LoRA Routing | | | | Efficiency | | | | |
| Existing systems | Training-free | Update | Access control | Selection | Mechanism | Gate size ($G$) | #parameters ($P$) | LoRA-scaling | Memory | Training Effort | Inference time | Task |
|---|---|---|---|---|---|---|---|---|---|---|---|---|
| SMoRA [35] | ✗ | ✗ | ✗ | Top-k | Rank | $N^2SD$ | $G×LY$ | $O(N)$ | $O(P)$ | $O(2^N)$ | $O(N)$ | ✎ |
| MoLE [18] | ✗ | ✗ | ✗ | Top-k | Seq | $N^2SD$ | $G×LY$ | $O(N)$ | $O(P)$ | $O(2^N)$ | $O(N)$ | 🖼,✎ |
| Diffusion-MoLE [36] | ✗ | ✗ | ✗ | Top-k | Seq + Token | $N^2SD+NSD$ | $G×LY$ | $O(N)$ | $O(P)$ | $O(2^N)$ | $O(N)$ | 🖼 |
| MoELoRA [19] | ✗ | ✗ | ✗ | Top-k | Token | $ND$ | $G×LY$ | $O(N)$ | $O(P)$ | $O(N)$ | $O(N)$ | ✎ |
| Retrieval-Augmented [20] | ✗ | ✗ | ✗ | Top-k | Token | $2kSD$ | $G×LY$ | $O(1)$ | $O(P)$ | $O(2^N)$ | $O(N)$ | ✎ |
| LLaVA-MoLE [37] | ✗ | ✗ | ✗ | Top-k | Token | $ND$ | $G×LY$ | $O(N)$ | $O(P)$ | $O(2^N)$ | $O(N)$ | 🖼,✎ |
| DynMoLE [38] | ✗ | ✗ | ✗ | Top-k,top-p | Token | $ND$ | $G×LY$ | $O(N)$ | $O(P)$ | $O(2^N)$ | $O(N)$ | ✎ |
| HDMoLE [39] | ✗ | ✗ | ✗ | Top-k, dynamic threshold | Token | $ND$ | $G×LY$ | $O(N)$ | $O(P)$ | $O(2^N)$ | $O(N)$ | 🔊 |
| LoRA-LEGO [40] | ✓ | ✗ | ✗ | Minimum semantic unit clustering | LoRA merge | None | None | $O(1)$ | $O(1)$ | $O(2^N)$ | $O(1)$ | ✎ |
| MiLoRA [41] | ✗ | ✗ | ✗ | Top-k | Seq | $NdN_{mod}$ | $NdN_{mod}$ | $O(1)$ | $O(1)$ | $O(2^N)$ | $O(N)$ | ✎ |
| MeteoRA [42] | ✗ | ✗ | ✗ | Top-k | Token | $ND$ | $G×LY$ | $O(N)$ | $O(P)$ | $O(2^N)$ | $O(N)$ | ✎ |
| SMEAR [43] | ✗ | ✗ | ✗ | Top-k | Seq | $ND$ | $G×LY$ | $O(N)$ | $O(P)$ | $O(2^N)$ | $O(N)$ | ✎ |
| LoraRetriever [44] | ✓ | ✗ | ✗ | Avg | Token | None | None | $O(1)$ | $O(1)$ | None | $O(N)$ | ✎ |
| **AC-LoRA (this paper)** | ✓ | ✓ | ✓ | Top-k | Seq | None | None | $O(1)$ | $O(1)$ | None | $O(N)$ | 🖼,✎ |

## 2 Motivation, Problem Statement, and Related Works

Besides documentation and code bases, corporate LLMs are trained with employee-specific data such as meeting records, emails/chat records on project progress, wiki entries, etc. Fig. 1 shows that information access typically follows the organization hierarchy. Users should only be able to access their data and the projects they participate in or manage. Naively, organizations can train separate models with non-overlapping sensitive documents. Maintaining these models is *prohibitively expensive*, as an organization with $n$ permission zones has $2^n$ distinct permission groups.

**Challenges with Single Foundation Model Training.** A single foundation model trained with all organizational data is easy to manage but poses security risks. LLMs retain some part of the training dataset, known as *memorization*, which can be reproduced or confirmed via membership inference attacks [24]. In practice, censorship methods are employed to monitor inputs and outputs to prevent sensitive data leaks. However, studies [25–27] show these mechanisms are often inadequate, as attackers can bypass them with jailbreaking [28, 29] and harmless-looking inputs [30–33]. Information leaks in corporate chatbots [1, 2, 34] threaten AI adoption in such contexts.

To show the real-world implications of memorization, we fine-tune LLAMA3.1-8B using two LoRAs on arXiv papers about confidential computing (CC) and quantum computing (QC), all published after the model's training cutoff. We evaluate *verbatim* memorization of the training dataset at inference time. Fig. 2 shows the information leakage (8, 12, 15, and 18 subsequent grams) from the training set. A large segment of text match (usually $\geq 12$ grams) is a telltale sign of memorization. We observe that memorization amplifies with a higher temperature ($T > 0$). We perform three inference runs (MP) at $T = 0.7$.

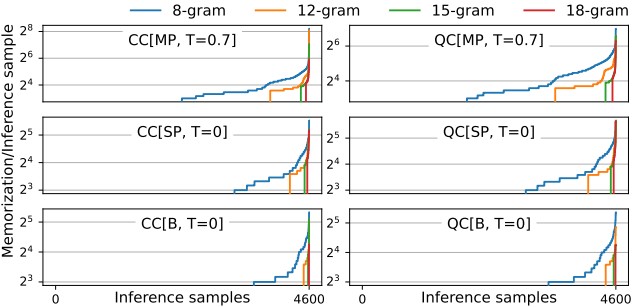

Figure 2: LLAMA-3.1 8B Memorization in multiple prediction (MP) with $T = 0.7$, single prediction (SP) and base model (B) with $T = 0$, on confidential computing (CC) & quantum computing (QC) datasets.

In our 12-gram experiment, the substring leakages from MP and SP above the base model are *in CC, 9.9k and 3.7k*, and *in QC, 15.6k and 4.7k* words. From this observation, we conclude that a single foundation model trained with all sensitive data is detrimental to maintaining information isolation and safety. Appx. A provides more details on the memorization experiment.

**Related Works and Drawbacks.** We discuss the drawbacks of several existing techniques.

1. Access-controlled RAG: RAG with access control [45] can allow or deny user access to a document. However, retrieving from external sources (storage, the internet) is expensive. Additionally, retrieving a set of large documents and putting them into context is detrimental to the performance of the LLMs, as shown by [5], as LLMs often fail to extract relevant information from long context. Too many documents can also lead to inferior performance [46]. Longer context significantly increases inference time and memory consumption. Recent research shows that RAG can produce unsafe LLM responses compared to the base model answer [11], fine-tune scores higher in accuracy compared to RAGs [47], and RAGs are ineffective in multi-hop queries [6].

2. Separate adapters for permission roles: Maintaining separate LoRAs for non-overlapping, *permissioned* datasets is feasible only if all users have a single permission role associated with a single dataset. Users of multiple permission zones, which reflect the organization's hierarchical structure, require fine-tuning new LoRAs with the merged datasets. However, this is prohibitively expensive as an exponential number of LoRAs ($n$ permission zones lead to $\mathcal{O}(2^n)$ LoRAs) is needed.

3. Training-free LoRA Mixing: There are two primary existing methods to combine multiple LoRAs without requiring any trained gating or routing mechanism. One approach is to mix the outputs of the LoRAs by averaging their results. Given the up and down projection of $n$ LoRAs as $A = \{A_1, \ldots, A_n\}$ and $B = \{B_1, \ldots, B_n\}$. The average output ($Y$) for input $x$ from the mixture of LoRAs is: $Y = \frac{1}{n}\sum_{i=i}^{n} B_i A_i x$. Another method produces a new LoRA with merged weights of the LoRAs as: $w_{\mathrm{merged}} = \frac{1}{n}\sum_{i=i}^{n} B_i A_i, Y = w_{\mathrm{merged}}x$. In both cases, the inference accuracy diminishes severely [44, 48] with increasing number of LoRAs.

4. Training-based LoRA Mixing: To improve the above techniques, MoLE [18] uses a trained gate to merge the entire output sequence from every MLP layer to combine tasks from every LoRA expert. The gate merges the output sequences based on the input encountered during the training phase. The gate parameter size increases linearly with the number of LoRAs and input sequence size. Like MoE [49], MoELoRA [19] utilizes sparse MoE activation with a trained gate. After every attention layer, the expert gate diverts a single token (unlike the sequence in MoLE) to an expert MLP. The routers/gates in MoE models act as load balancers, trained jointly on all experts' data, which risks including confidential information, even if an expert is disabled due to the permission set. Alternatively, the routers can be trained for every possible permission set ($\mathcal{O}(2^n)$), bringing us back to the same challenge of training as many LoRAs. This shows that MoE-style LoRA mixing techniques are *not* directly suitable for strict access control.

**Threat Model.** AC-LORA assumes that the attacker can remotely access the LLM chatbot and send unlimited queries, aiming to maximize the retrieval of unauthorized information. They can inject arbitrary system commands or special tokens into queries and modify documents they can write, like personal records (corporate email, chat accounts, meeting recordings, etc.), and project data. We also assume the attacker cannot steal the identity to impersonate a user.

**Requirements.** Given the above-mentioned problem space, we summarize the following requirements for a secure corporate AI chatbot with strict information access control:

→*RQ 1*: *Strict access control policy.* A user without the proper access rights cannot access restricted information or bypass the access control through means such as prompt injection.

→*RQ 2*: *Arbitrary permission rules.* The model can handle users' requests with new permission rules never encountered before, while maintaining the access control policy.

→*RQ 3*: *Efficient update.* Information can be added, updated, or deleted with minimal effort.

→*RQ 4*: *Efficiency.* Ensuring that all of the above changes can be addressed without adding a significant number of parameters to the model to avoid a significant increase in inference latency.

## 3 AC-LORA: Permission-Aware LoRA Retrieval and Mixing

We present AC-LORA: an end-to-end access control-aware LLM inference system. It integrates LoRA-based retrieval with dynamic LoRA mixing to efficiently support an exponential number of permission rules, while ensuring users can access all information to which they are authorized.

**Main Observation.** AC-LORA finetunes and maintains separate LoRAs for different permission zones. We assume a permission zone consists of projects or topics. We use three open-source datasets: Flan-v2 [21], RepLiQA [22] for text, and wikiart [23] for multi-modal. Figs. B.1 to B.3 show that topic embedded spaces are separable. Tasks such as anli_r1 and anli_r2 in the Flan-V2

(Fig. B.1) are variants of the same task, and their embeddings overlap. This observation is further reinforced by the pairwise cosine similarity score depicted in Figs. B.4 and B.5.

**Isolated LoRA Fine-tuning and Knowledge Injection.** Our memorization observation (cf. Sec. 2) suggests that ensuring information isolation between different permissioned data zones requires fine-tuning on individual, isolated datasets with separate LoRAs (RQ. 1).

The base LLM contains public knowledge, while a specific LoRA(s) is required for the base model when a user queries a restricted topic. We fine-tune 17 LoRAs using the RepLiQA [22] dataset, which contains a human-evaluated *mock news dataset* that our base model, LLAMA-3.1-8B, has never seen during its training. We use GEMMA-3-27B to grade the responses (on a scale of $[0, 5]$, cf. Appx. D.3.2). Fig. 3(a) shows the grade of Cybersecurity topic from the RepLiQA over different ranks ($r$) and finetune steps. We observe that LoRAs can reliably inject domain-specific knowledge into the base model, assuming the base model has not been trained on a similar dataset. We observe that the most crucial factor is the dataset size; a smaller dataset can lead to model overfitting. Across ranks, LoRAs perform the best at a fine-tune step size of ∼220. Fig. 3(b) shows a summarized result for all RepLiQA datasets, and it shows that the finetuned model performs *better in every subject* than the base model with an average of

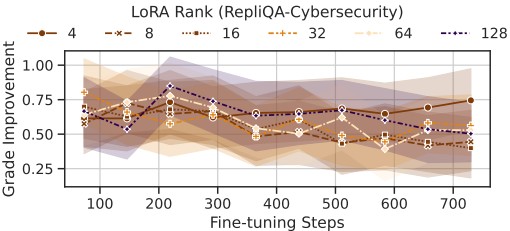

(a) RepLiQA: Cybersecurity topic

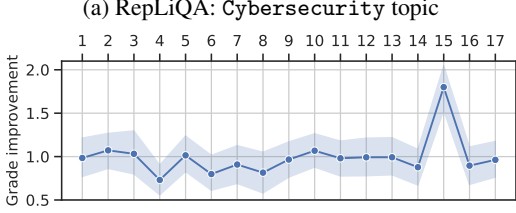

(b) RepLiQA: all topics (x-axis: Tab. C.2).

Figure 3: Knowledge injection of RepLiQA (split 0) dataset in LLAMA-3.1-8B.

0.959 grade improvement. Our observation aligns with existing works [13], which evaluate the effect of knowledge injection using LoRA.

**Secure LoRA Retrieval and Similarity-based LoRA Merging.** AC-LORA maintains two vector databases. LORA-DOC EMBED contains the mapping between the LoRA and their corresponding fine-tuned document embeddings (chunked in 100 tokens). LORA-PERMISSION contains the permission information of the users. Each user (uniquely identified by their User-ID attribute) is associated with an $n$-dimensional ($n$ LoRAs) vector, where the vector elements denote deny (0) or access (1) to a specific LoRA. Given a tuple: {query, User-ID} from the user, AC-LORA retrieves a set of candidate LoRAs based on the cosine similarity between the embeddings of the query and the training dataset. We denote the set of candidate LoRAs along with their cosine similarity scores as $\mathcal{O}$. The User-ID retrieves the permission vector from LORA-PERMISSION. We denote the set of permissible LoRAs of the given user as $\mathcal{P}$. AC-LORA retrieves and loads the set of relevant LoRAs $\mathcal{L} = \mathcal{O} \cap \mathcal{P}$ from LORA-DOC EMBED. The similarity scores of the LoRAs in $\mathcal{L}$ are also passed to the mix-gate after each MLP layer. Given the LoRAs in $\mathcal{L} = \{L_1, L_2, \ldots L_k\}$ where $k \leq n$ and their corresponding normalized similarity score $\{S_1, S_2, \ldots, S_k\}$, such that $\sum_{i=1}^{k} S_i = 1$, then for a query $\mathcal{Q}$ the output for each LoRA in each layer $L_i \in \mathcal{L}$ is $y_i = Q(A_i B_i)$ ($L_i$'s low rank components: $A_i, B_i$). The final output for each layer is then $\mathcal{Y} = W + \sum_{i}^{k} S_i y_i$ where $W$ is the base model weight. Note that the mixing is completely *training-free*, i.e., the model owner does not need to retrain it for every possible combination of LoRAs (RQ. 2), and therefore enables faster and memory-efficient inference (RQ. 4) due to the absence of gate parameters. Not loading any LoRAs outside the user's permission also ensures that no sensitive data is leaked (RQ. 1).

**Combining Knowledge from LoRAs.** Fig. 1 shows that corporate AI chatbots should be able to combine information from different datasets. The example query *"How many cores are in the CPU?"* might have a different answer depending, for example, on the platform or generation. Therefore, it is important to collect and combine the relevant information across different permission zones (given the user has the correct access rights). AC-LORA's similarity-based LoRA merging captures domain-specific knowledge across non-overlapping permission zones. This knowledge combination enables AC-LORA to avoid maintaining all possible permission zone LoRAs (RQ. 2).

**Answer Hinting.** The hint set are determined as $\mathcal{H} = \mathcal{O} - \mathcal{L} = \mathcal{O} - (\mathcal{O} \cap \mathcal{P})$. The metadata of the LoRAs in $\mathcal{H}$ are retrieved from LORA-DOC EMBED and given to the user as a hint that there might be better answers given the queries and how to apply permission for them. This acts as valuable guidance for the users to apply for the correct permission to further refine their response. A curious/malicious user can gain knowledge of the possible existence of the information. Disabling the hint for sensitive LoRAs (e.g., specification of an upcoming product) will prevent AC-LORA from mentioning the existence of a specific dataset to not-permitted users.

**Update Operations.** Unlike the majority of existing works on LoRA mixing (see Tab. 1), AC-LORA is more flexible. To remove a dataset, the model owner must only remove one entry ($\mathcal{O}(1)$ operation) from both the LORA-DOC EMBED and LORA-PERMISSION databases. To modify an existing permission zone (add/delete/modify), the model owner needs to fine-tune the specific LoRA with updated data, recompute the embedding of the fine-tune dataset, and update the LORA-DOC EMBED vector-DB with the updated LoRA and the document embeddings. This does not affect the LoRA mixer, as it is only dependent on the individual cosine similarity score of the query and the document embedding vectors (RQ. 3). Updating the access control vector of the user only requires updating one entry in LORA-PERMISSION, which is also an $\mathcal{O}(1)$ operation.

**Summary of the Secure LoRA Retrieval and Merging.** The step-by-step process of our proposed system AC-LORA, depicted in Fig. 4 are: ① The user passes their query and credential information to the system. First, the query goes to an embedding model to produce a vector embedding. This embedding is then passed to LORA-DOC EMBED for a top-k similarity search. ② The top-k similarity search produces $\mathcal{O}$: top-k LoRAs along with their cosine similarity scores with the user query. ③ The user permission passes as the input to the LORA-PERMISSION that retrieves the set of permitted LoRAs. ④ The permitted LoRAs are then passed to the base model. ⑤ The outputs of the LoRAs are mixed with the same proportion of the similarity score of $\mathcal{O}$. ⑥ The merged model outputs the main `Response`, which abides by the strict access control policy in the LORA-PERMISSION. ⑦ The `Hint` is derived from $\mathcal{O}$ based on the non-permitted LoRAs with a higher similarity score.

**AC-LORA scaling with number of documents.** AC-LORA is independent of the number of relevant LoRAs. We can summarize the effects into two scenarios: (I) If all relevant documents are from the same permission domain (e.g., one LoRA), we choose to assign the average similarity score of the documents to that specific LoRA. (II) If the documents are spread over many permission domains, we assign the average similarity score of the relevant document to each permission zone, corresponding to its respective LoRA. In both cases, AC-LORA only calculates the similarity score for the permission domain if the user has access rights to it. Therefore, unlike RAG, AC-LORA does not suffer from lower performance (due to increased context size) when retrieving many relevant documents. However, in a rare scenario, a query from a very high-privilege user (e.g., a CEO) may require many LoRAs simultaneously. In such a case, the GPU memory will be a bottleneck, and can be mitigated by using a suitable parallelization strategy, such as pipeline parallelization.

**Multi-modalities.** AC-LORA extends beyond text-based models. *Similar to* the text, a tight access control mechanism is applied to such multi-modal scenarios. Existing work [36] utilizes a mixing of LoRAs on stable diffusion to enhance the overall image quality when using LoRAs specialized on partial human features. AC-LORA uses a similar mechanism to train isolated multi-modal LoRAs based on QWEN2-VT and stable diffusion model: STABLE-DIFFUSION-V1-4.

## 4 AC-LORA Evaluation

This section describes AC-LORA's end-to-end evaluation. We run our experiments on two workstation GPUs with 48GB GDDR6 VRAM. Additional details regarding the setup and implementation can be found in Appx. E. This section summarizes the key results of the AC-LORA evaluation. Further results are discussed in Appx. C.

### 4.1 Methodology

**Datasets.** In the following, we describe our setup for three different datasets:

1. RepLiQA: RepLiQA (split 0 cf. Fig. B.2) consists of several small artificial articles covering various topics, along with multiple question-answer pairs per article. We split it into an 80-20 training-test set, ensuring with stratification that each article is seen at least once in the training

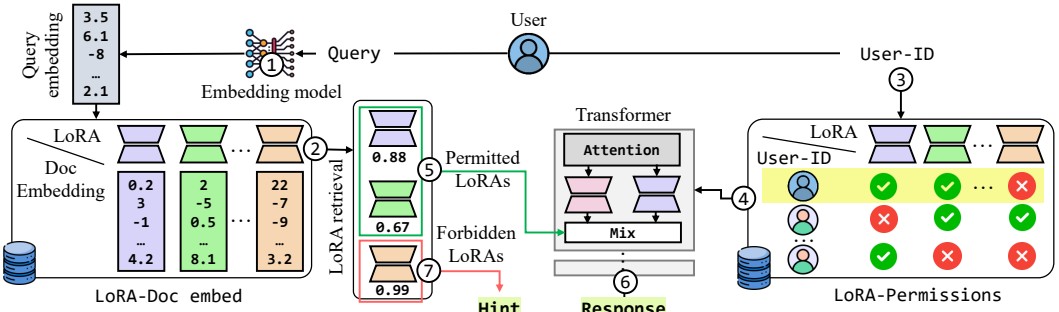

Figure 4: A high-level overview of AC-LORA showing LORA-DOC EMBED and LORA-PERMISSION vectorDBs. LoRAs are retrieved and mixed with the base model based on the user's query embedding and permission. Relevant and non-permitted LoRAs can be returned as a hint.

set. We finetune 17 LoRAs (with rank $r = \alpha = 64$), one for each topic, using LLAMA3.1-8B-INSTRUCT as the base model. As seen in Fig. 3(a), we keep the finetune step size $\sim 200$ to avoid overfitting. The training set comprises four data points per question: two with the document and two without, each pair once with a short and long answer. Including question-only pairs improved results as it aligns with the test set. To build the embedding database for AC-LORA, we use the ALL-MNET-BASE-V2 [50] sentence transformer and split the training set for each LoRA into chunks of 100 tokens, adding the corresponding LoRA as a tag.

2. Flan: FlanV2 contains datasets of 10 task domains (cf. Tab. 2). We use the identical setup of [44], including the LoRAs shared with parameters ($r = 8$ and $\alpha = 16$). We utilize their test set, which consists of 50 data points per task. As the training set used for the different LoRAs was not shared, we constructed one based on the official FlanV2 dataset for the retriever. In particular, we take the first 30k (or fewer for smaller tasks) samples of each selected task as the training set and build the database as described above. As in LoRARetriver, we use the BLEU score to evaluate the translation, ROUGE for the STRUCT-TO-TEXT TASKS, and EXACT MATCH for the rest.

3. WikiArts: We query QWEN2-VL to generate descriptions of the images from the wikiarts [23] dataset (see Appx. D) to construct the text-embedding for the retrieval. We then finetune STABLE-DIFFUSION-V1-4 with the images to generate 27 LoRAs separated by the *style* attribute.

**Combining Knowledge.** Combining different LoRAs across different permission zones is important for AC-LORA(Cf. Sec. 3). Although existing works [18, 20] show that combining LoRAs can be used to combine tasks from different LoRAs, e.g., translating from English to Spanish and then from Spanish to German, to answer queries for English to German. However, to our knowledge, no existing work shows that combining different LoRAs can increase a model's information recall. Retrieving more than just a single (best-fitting) LoRA and combining them introduces new information and increases the response quality. To demonstrate, we create a dataset (CS-COMBI) from Cybersecurity News category in RepLiQA. For each text, we ask a reasoning model (DEEPSEEK-R1-32B) to extract the most (between 3 and 12) relevant facts. We then divide these facts randomly into two groups. From these two groups, we generate:

1. Context: We ask the model to write a text that exclusively includes the facts given, creating two new articles with parts of the information missing.

2. Combined QA Pairs: Taking one fact from the first group and one from the second, we ask the model to generate a question and answer pair that requires both facts to answer.

3. Single QA Pairs: Taking two facts from the same groups, we ask the model to generate a question and answer pair that requires both facts to answer.

We built one test and two disjoint training sets (with and without context), each containing at least one question for each context in its corresponding group. We then fine-tune LLAMA3.1-8B-INSTRUCT for three epochs and produce two LoRAs on these two training sets using different $r (= \alpha) \in \{4, 8, 16, 32, 64, 128\}$. The prompts for extracting facts and additional details are in Appx. D.

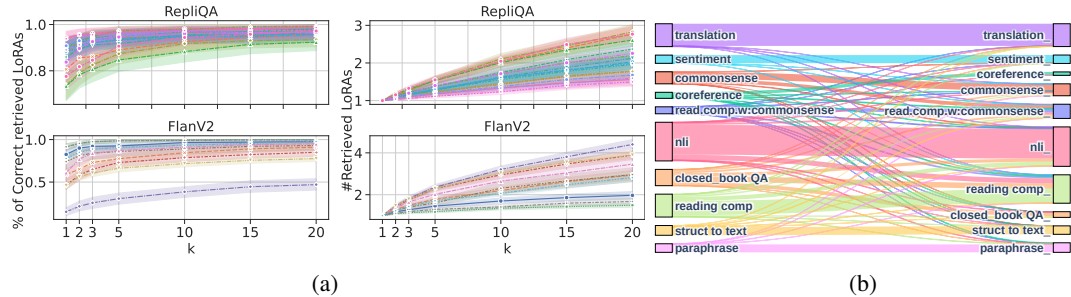

Figure 5: (a) LoRA retrieval performance in RepLiQA and FlanV2 for different top-k. Left: % of queries per field for which the correct LoRA ∈ the set of retrieved LoRAs. Right: # retrieved LoRAs. (b) Retrieval of FlanV2 target domains (left) and corresponding retrieved domains (right).

Table 2: AC-LORA evaluation on FlanV2 dataset and comparison with other LoRA approaches. The baselines are extracted from LoRARetriever [44] for comparison. The best result is **bold**, while the second best is underlined.

| Task | Perfect Selection | Selection | | Fusion | | Mixture | | MoE Top1 | MoE Top3 | MoE Soft | AC-LORA (95%-CI) (k=3, fetch_k=10) |
|------|------|------|------|------|------|------|------|------|------|------|------|
| | | IID+ | OOD* | IID+ | ODD* | IID+ | OOD* | | | | |
| Struct to text Rouge-1 | 64.0 | 61.3 | 50.1 | 49.4 | 45.9 | 55.9 | 50.4 | 45.6 | 46.8 | 47.9 | **61.7** (58.2-64.7) |
| Struct to text Rouge-2 | 39.6 | **37.0** | 26.6 | 25.7 | 23.5 | 30.0 | 26.4 | 21.9 | 22.9 | 23.8 | **37.0** (33.8-40.3) |
| Struct to text Rouge-l | 57.0 | 54.5 | 43.9 | 43.6 | 40.3 | 49.5 | 44.0 | 39.8 | 40.7 | 41.7 | 54.3 (51.1-57.3) |
| Translation | 13.1 | 12.8 | 12.0 | 12.2 | 12.3 | 12.8 | 12.2 | 9.5 | 10.5 | 10.7 | **13.6** (11.37-15.45) |
| Commonsense | 62.5 | 55.5 | 46.0 | 51.0 | 48.0 | 61.5 | 50.0 | 54.5 | 52.0 | 51.5 | **65.0** (58.5-72.0) |
| Sentiment | 90.0 | 89.5 | 89.0 | 79.0 | 78.5 | 89.5 | **90.5** | 70.0 | 75.0 | 74.5 | 90.0 (86.0-94.0) |
| Reading Comp | 67.3 | 51.7 | 40.3 | 47.3 | 45.0 | 51.3 | 47.3 | 48.7 | 47.7 | 48.7 | 55.3 (49.0-60.6) |
| Closed book QA | 45.0 | 40.0 | 43.0 | 41.0 | 37.5 | 45.0 | **48.5** | 40.5 | 38.5 | 40.0 | 39.0 (30.5-43.0) |
| Coreference | 52.0 | 50.0 | 46.0 | 47.0 | 53.0 | **63.0** | 49.0 | 61.0 | 59.0 | 57.0 | 54.0 (44.0-64.0) |
| Read.comp.w/commonsense | 69.0 | **69.0** | 30.0 | 35.0 | 19.0 | 46.0 | 40.0 | 31.0 | 29.0 | 29.0 | 63.0 (49.0-69.0) |
| Paraphrase | 65.5 | 58.0 | 45.5 | 45.5 | 44.0 | 56.5 | 45.5 | 42.0 | 38.5 | 36.0 | **63.0** (56.4-69.5) |
| NLI | 72.3 | **70.0** | 60.6 | 51.4 | 53.8 | 67.9 | 64.3 | 50.3 | 49.6 | 48.3 | 68.7 (64.2-72.5) |

+IID: access any LoRA for every test sample, encompassing the LoRA specific to the sample's task. *OOD: for each test sample, the LoRA associated with its specific task during the retrieval phase is masked.

## 4.2 Main Results

**Retriever Performance.** We now discuss our two main results: first, we demonstrate that our retriever achieves high accuracy in retrieving the correct LoRA when given a query. As depicted in Fig. 5(a), with increasing $k$ (for the top-$k$ retrieved documents), the accuracy of having the correct LoRA in the set of retrieved LoRAs approaches one, while keeping the number of retrieved LoRAs under 3 for RepLiQA and 5 for Flan. Fig. 5(b) confirms this by displaying the connection between a query from a domain (left) and the retrieved LoRA domains (right) that answer the query (in FlanV2). More detailed results and discussion are provided in Appx. C for both FlanV2 and RepLiQA.

**Inference Results.** We now provide AC-LORA's inference results. 1. RepLiQA: During inference, AC-LORA retrieves the relevant LoRAs ($k = 10$) and mixes them based on the cosine similarity with the query. Fig. 6(a) shows the AC-LORA's mixed LoRA inference grades (judged by GEMMA-3-27B model), compared to the single finetuned LoRAs (perfect selection) specific for the given query.

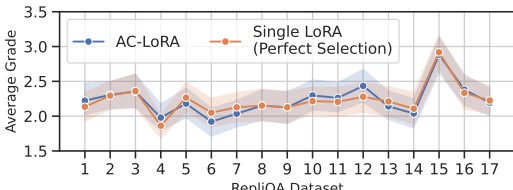

(a) AC-LORA inference grades w.r.t to single LoRAs.

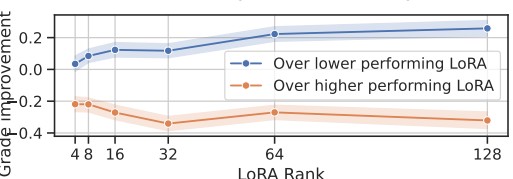

(b) Mixed LoRAs grade improvement on CS-COMBI.

Figure 6: AC-LORA evaluation on RepLiQA.

AC-LORA performs very close to the perfect selection on most topics, and in some, it exceeds the perfect selection due to mixing with relevant LoRAs from other domains.

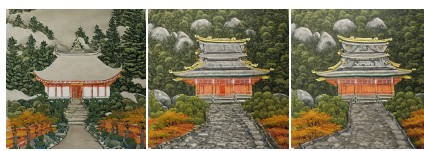

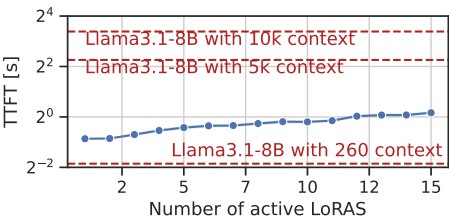

Figure 7: AC-LoRA multi-modal LoRA mixing on Wikiart dataset. (Prompt 8)

Figure 8: AC-LoRA's time-to-first-token latency (TTFT) in 90 input-tokens on varying # LoRAs.

2. FlanV2: Unlike RepLiQA, for Flan, we follow similar accuracy metrics as state-of-the-art Lo-RARetriever [44] to evaluate AC-LoRA. Tab. 2 shows that AC-LoRA matches or exceeds Lo-RARetriever's accuracy *without* requiring any optimization or training for LoRA mixing. More details are in Appx. C. FlanV2 focuses on different formats (tasks) rather than information. Therefore, AC-LoRA also effectively isolates tasks.

3. Multi-modal: Fig. 7 highlights AC-LoRA multi-modal performance where images are generated using the prompt in Prompt 8 with increasing top-$k$ ($k \in \{1, 2, 3\}$) and using (in the retrieval order) the LoRAs of *Ukiyo_e*, *Impressionism*, and *Symbolism*. Additional multi-modal results are in Appx. C.4.1.

**AC-LoRA effectiveness on overlapping knowledge groups.** AC-LoRA utilizes an off-the-shelf embedding model to calculate the embedding space and the similarity score. For datasets with overlapping permission domains, a fine-tuned embedding model can ensure separation. Overlapping knowledge groups are not detrimental to AC-LoRA. If one or more overlapping datasets are not included in the user's permitted list, the specific LoRA(s) will not be used to answer the question. If one or more are permitted, they will be merged based on the similarity score. Anli_r1,r2,r3 in FlanV2 are very similar and have a large overlapping embedding space.

**Combining Knowledge.** We fine-tune (cf. sec. 4.1) two LoRAs on disjoint datasets. While a single LoRA can answer some test questions, most require information from both. In Fig. 6(b), we illustrate, in blue, the average improvement in answers using both LoRAs compared to the lower-scoring LoRA, and in orange, the improvement over the higher-scoring LoRA. Although combining LoRAs improves performance over the weaker one, it generally performs worse than the higher-performing LoRA. The query still requires both LoRAs to answer, but not with the same weight, leading the LoRA with less information on the subject to introduce noise. We observe 7.71% of test queries show improvements over both, and depict a similar behavior to the one described in Fig. 1. Specific examples of such query-response pairs are provided in Appx. C.2.2.

**Effect on Inference Latency.** Fig. 8 shows the time to first token generation latency with an increasing number of active LoRAs. We construct a prompt (90 input tokens long) such that with increasing $k$, we can retrieve an increasing number of LoRAs (RepLiQA). As a comparison, we also provide vanilla LLAMA3.1-8B's latency with 260, 5K, and 10K context sizes to visualize the effect of an equivalent RAG-like solution that retrieves and sets the entire relevant documents to the context. This shows AC-LoRA is efficient and satisfies RQ. 4.

**Effect on memory.** Loading the 8B base model with 17 LoRAs ($r = 64$) requires ∼40GB of memory without quantization. Similarly, Flan with 48 LoRAs ($r = 8$) uses only slightly more and still fits on a single 48GB GPU. These requirements depend on the base model size, LoRA configuration (targeted modules, number of layers, and rank), and the prompt (and context) length. However, parallelization techniques such as pipeline parallelization can allow AC-LoRA to load more LoRAs. Additionally, in a memory-constrained environment, one could keep frequently used LoRAs on the GPU while loading less common ones on demand, trading off a slight latency increase.

# 5 Discussions

**Societal impacts.** There are other scenarios where AC-LoRA can enforce strict access control while maintaining high inference quality, besides corporate AI chatbots.

1. Safeguard users from unsafe content (e.g., illegal advice or violent images): One can isolate the training sets (of the said contents) and finetune separate LoRAs, which could, for example, only be accessed by authorized personnel (e.g., law enforcement).

2. Foundation models with IP data: As recent reports [51] indicate, unlawful usage of IP data in training may have severe legal implications; AC-LORA could allow using such data by keeping it on licensed LoRAs and loading it with the base model for specific users. However, such use cases require further investigation, and the details of how such systems could be implemented using AC-LORA are out of the scope of this paper.

3. Bypass content filter: If the content filtering is implemented by model alignment [52], there are three scenarios in which they could be affected by AC-LORA:

- *No LoRAs are retrieved:* If the user query does not match any LoRA, none will be retrieved, and the base model will answer the query. Therefore, the original alignment of the model remains unchanged.
- *LoRA(s) are retrieved:* It is challenging to ensure model alignment without considering this during fine-tuning.
- *Specific LoRAs for bypassing filters:* A set of LoRAs could be specifically fine-tuned to bypass the content filter (e.g., LoRAs used by law enforcement), making the bypass of the filter intentional.

However, the broader impact of such a system is complex, requires further analysis, and is beyond the scope of this paper.

**Limitations.** We now discuss some of the limitations of AC-LORA.

1. General Limitation of LLMs and finetuning: LLMs perform well on some tasks but have notable shortcomings like hallucinations, context scaling issues, and limited reasoning abilities. Reasoning models help address some gaps, like multi-hop reasoning, but major issues persist. Importantly, AC-LORA relies on LLMs' capacity to learn and integrate new data during fine-tuning, making its design agnostic to future advancements in reasoning models, as it is applicable on top.

2. Hinting: The hinting mechanism in Sec. 3 can introduce new attack vectors. Although useful in specific scenarios, it risks membership-inference-like attacks that could expose confidential data. We recommend using it cautiously, ideally with LoRAs on non-sensitive datasets.

3. Combining Knowledge: While we have presented a first experiment and dataset suggesting that different LoRAs can combine knowledge, a more extensive analysis is required to better understand the extension (and limitation) of such capabilities.

4. Frequent Swapping of LoRAs: We assume that either all LoRAs fit on the devices, or LoRA swapping between inference rounds is minimal. If not, the time to first token increases significantly, as shown in Fig. C.9. However, we believe this assumption is reasonable. In the improbable worst-case scenario, where multiple (or all) LoRAs must be loaded in each inference round, performance could, for example, be improved by reducing the value of $k$ or optimizing the batching algorithm.

5. Multi-Modal: Our evaluation on other modalities serves as a proof of concept rather than a comprehensive analysis. A thorough assessment would require more resources, including a new dataset and more robust evaluation methods, which are beyond the scope of this work (see Appx. C.4).

## 6  Conclusion

In this paper, we propose AC-LORA, a multi-modal, access-control aware LoRA serving system that requires no additional training for mixing responses by different LoRAs. AC-LORA is efficient, can retrieve and mix relevant LoRAs based on the user's query, while maintaining strict organization information access control policies. AC-LORA evaluation shows that deploying and providing high-quality responses is practical.

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

# Appendix

## Table of Contents

## A   LLM Memorization Evaluation

We construct an experimental pipeline consisting of several stages: preparing the dataset, fine-tuning the base model, performing inference, and comparing the model's predictions against the training data. In the following sections, we provide a detailed description of each step in the pipeline.

### A.1   Dataset Preparation

To create the datasets, we use the arXiv API to download research papers on two topics: confidential computing (CC) and quantum computing (QC). The specific URLs used to retrieve the papers are listed in Tab. A.1.

| Topic | arXiv API URL (documentation: https://info.arxiv.org/help/api/basics.html#using) |
|---|---|
| Confidential Computing | https://export.arxiv.org/api/query?search_query=all:confidential+AND+all:computing&max_results=500 |
| Quantum Computing | https://export.arxiv.org/api/query?search_query=all:quantum+AND+all:computing&max_results=500 |

Table A.1: arXiv API URLs used for data retrieval

After downloading the papers, we filter out the ones published before 2024. Then, we convert the PDF files into plain text and split the resulting text into smaller chunks. Each chunk is then input into the LLAMA3.1-8B model, accompanied by a prompt instructing it to generate five question-answer pairs based on the given text as context:

> USER: Write the questions and corresponding answers, and do not repeat the given context or any final answer. Generate five questions and their corresponding answers from the given context. {context}

The final dataset comprises 15,459 question-answer pairs related to confidential computing and 15,466 question-answer pairs related to quantum computing. Finally, both datasets are partitioned into training and test sets using an 80-20 split.

## A.2 Fine-Tuning

The second step of the experimental pipeline involves fine-tuning an LLM for text generation using LoRA. We follow this approach to assess the extent to which LoRA adapters memorize training data, i.e., to evaluate how much of the original input is retained within the adapted parameters during fine-tuning. We load the base model, LLAMA3.1-8B, using 4-bit precision, nested (double) quantization, with normalized 4-bit quantization type and bfloat16 as the compute data type. We configure the LoRA adapters with an attention dimension $r = 16$, scaling factor $\alpha = 64$, and a dropout probability 0.1.

For the training itself, we adopt the same hyperparameter configuration used in the Stanford Alpaca project [53], due to the similarity between our datasets and those used in Alpaca — both in terms of size and structural format (instruction-answer pairs). This also helps to rule out overfitting as a contributing factor to memorization. We also use the same prompt format as in Alpaca.

Fine-tuning was performed using the same experimental setup described in Sec. 4.

## A.3 Inference

In the inference phase, we combine the individual LoRA adapters with the base model and prompt the fine-tuned models using inputs from the test dataset. In a real-world scenario, the model will likely encounter prompts that resemble those from the training set. Therefore, we use the test set, which shares a similar context with the train set since they originate from the same source, but differ enough to simulate real-world conditions. Our objective is to evaluate the model's memorization after fine-tuning, without having direct access to the training set, while still using similar prompts. We repeat the experiment in three distinct variants.

In the first variant, we apply the greedy search decoding strategy to obtain a single deterministic prediction per query. These predictions are consistent and can always be reproduced.

In real-world scenarios, attackers can prompt a model as often as they like. Consequently, repeated prompting can increase the likelihood of the model revealing memorized content, amplifying the risk of information leakage. We adopt a second experimental variant using the multinomial sampling decoding strategy to reflect this threat model. Specifically, we modify the default LLAMA3.1-8B generation configuration by slightly increasing the temperature from 0.6 to 0.7, and setting the top_p parameter to 1.0 instead of 0.9. This approach enforces more diverse predictions. We prompt the model three times, generating multiple prediction candidates.

In the third variant, we apply the greedy search decoding strategy again, but this time using only the base LLAMA3.1-8B model, without combining it with any LoRA adapters. This helps isolate the contribution of the LoRA adapters, allowing us to assess how much newly introduced knowledge is memorized by the adapters versus what the base model retains.

## A.4 Prediction Evaluation

The final stage involves a quantitative measurement of LLM memorization by comparing each generated prediction from the prediction set $P$ against each entry in the corresponding model's training dataset $S$. Unlike previous work that relies on concepts such as eidetic memory [54] and adversarial compression [55] to define and measure LLM memorization, our work aims to quantify memorization using simple string comparison techniques directly.

We compare each prediction $p \in P$ against each question-answer pair $s \in S$ from the corresponding model's training dataset by searching for all the common substrings between $p$ and $s$. Importantly, we treat $p$ and $s$ as sequences of words (rather than characters), where tokens are defined by whites-

Figure A.1: An example of a prediction generated by the quantum computing LoRA using the greedy search decoding strategy (left) and training set entries whose segments are contained within the prediction (right). The highlighted text indicates matching sequences with length greater than or equal to $n = 8$ words. The prediction has an absolute memorization score of 43 and a relative score of 0.387.

pace separation. We further enforce a minimum substring length $n$, measured in consecutive words, to ensure that only meaningful overlaps are considered in the analysis. We repeat the experiment for $n \in \{8, 12, 15, 18\}$.

For this purpose, we generalize the Longest Common Substring (LCS) Suffix Tree algorithm [56], to search not only for the longest common substring, but also for all common substrings between two strings [57]. We additionally adapt the algorithm to include only the substrings of length greater than or equal to $n$ words.

This process results in a set of $|S|$ overlapping intervals for each prediction $p$, where each range corresponds to the overlap with a specific training example $s$. To quantify memorization, we aggregate the intervals across all $s \in S$ to compute a global overlapping interval — the union of all sequences within $p$ that are directly and exactly memorized from the training set. We then compute two memorization scores for each prediction: an absolute score, defined as the total number of memorized words within the global interval, and a relative score, calculated as the ratio of captured words to the total number of words in the prediction. Fig. A.1 shows an example of a prediction alongside training set entries whose segments are memorized verbatim. In the case of multinomial sampling, where we generate three predictions per test query, we additionally aggregate the global intervals from all three predictions. We avoid double-counting when merging the intervals, such as when a captured substring from one prediction is partially or entirely contained within a longer overlapping substring from another. We then compute the cumulative absolute score based on the total number of memorized words within the merged interval.

It is important to note that finding all common substrings of two strings is prohibitively expensive, with time complexity of $\mathcal{O}(m + n)$, where m and n are the lengths of the two strings. Due to the number of experiments, the size of the training and test datasets, and the lengths of the generated predictions, the comparison stage required significant computational resources. The whole process took over 15 days to complete when executed in parallel across seven nodes.

## B  Dataset Embedding

Figs. B.1 to B.3 show the embedding space for FlanV2, RepliQA, and wikiart dataset respectively. As mentioned previously, ALL-MNET-BASE-V2 [50] was used to generate the dataset embeddings.

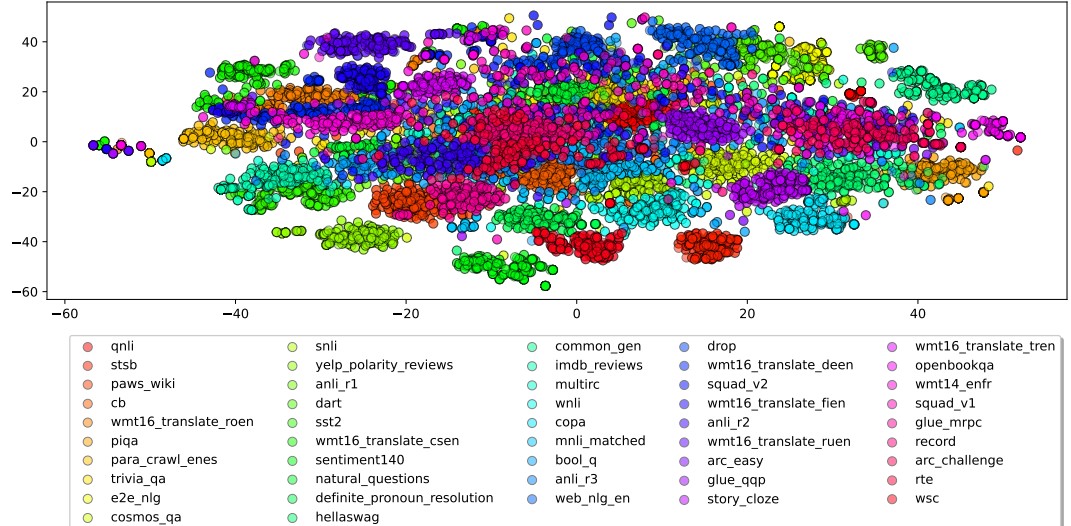

Figure B.1: Embedding space of the Flan-v2 [21] Dataset.

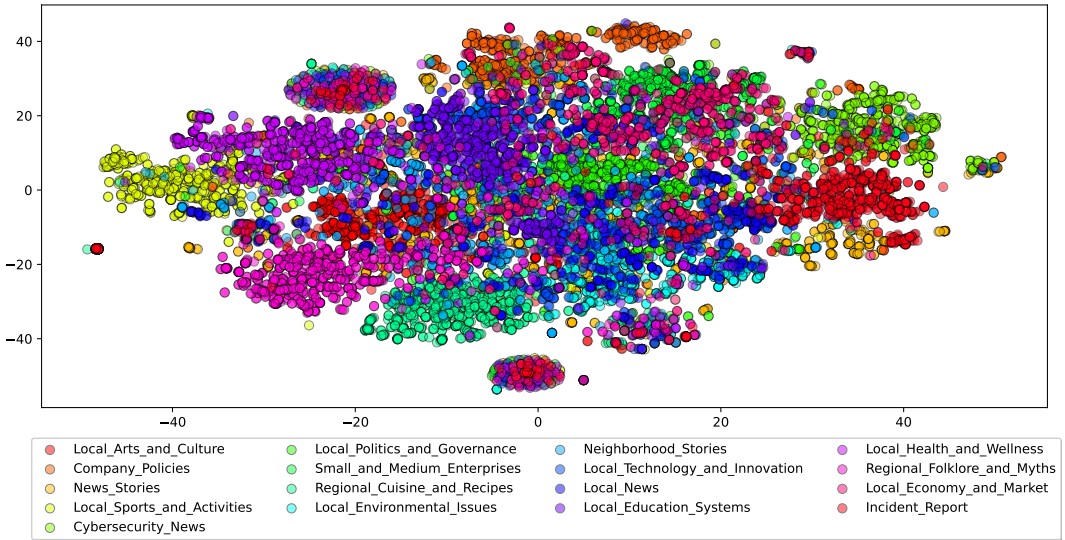

Figure B.2: Embedding space of the RepliQA [22] Dataset.

Appx. B and fig. B.5 additionally show the pairwise cosine similarity of these three datasets. To calculate the cosine similarities between each pair of topics in a dataset, we first calculate the centroid of all document embeddings in a topic. This embedding centroid is then used as the representative embedding for the specific topic and is used to derive the pair-wise cosine similarity.

## C  Additional AC-LoRA Results

In the following subsection, we will provide more detailed results from AC-LoRA evaluation.

### C.1  Flan

In Tab. C.1 we showcase the full comparison to [44] on Flan-v2. Similarly to the briefer version, we can see that AC-LoRA matches or outperforms other methods in most tasks. Unlike the results on RepLiQA, if the wrong LoRA is retrieved, the output format will (in some cases, drastically) change and thus receive a worse exact match score, even if the content of the answer is correct. Given the nature of AC-LoRA, it is unsurprising that its performance is poorer on tasks evaluated by exact

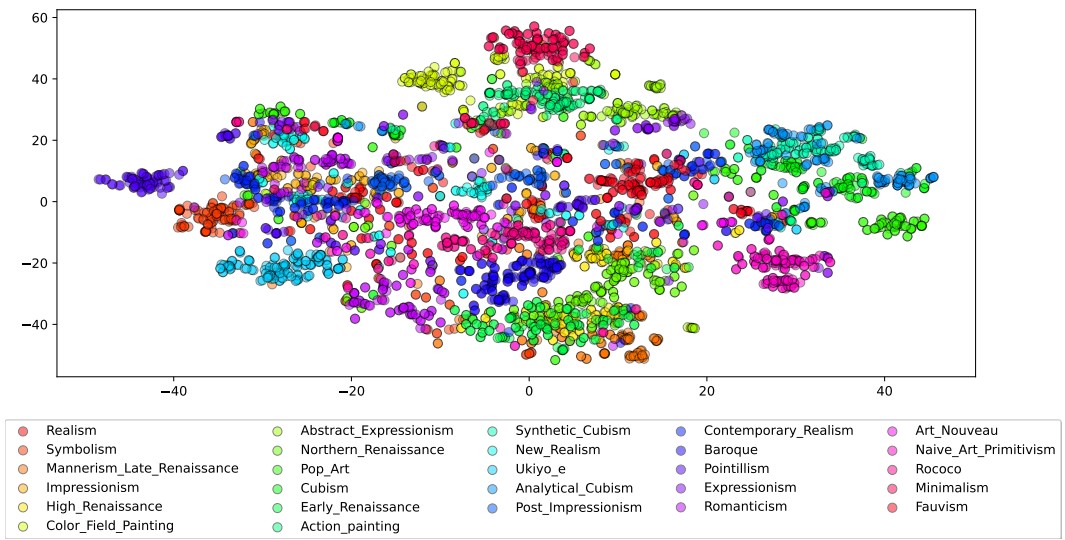

Figure B.3: Embedding space of the generated Wikiart prompts.

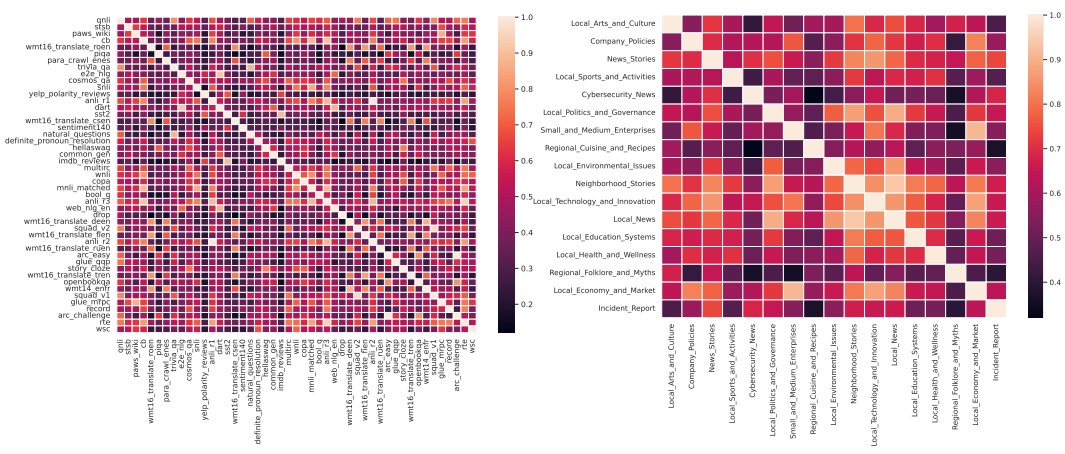

Figure B.4: All pair cosine similarities of Flan-v2 [21] and RepliQA [22] datasets.

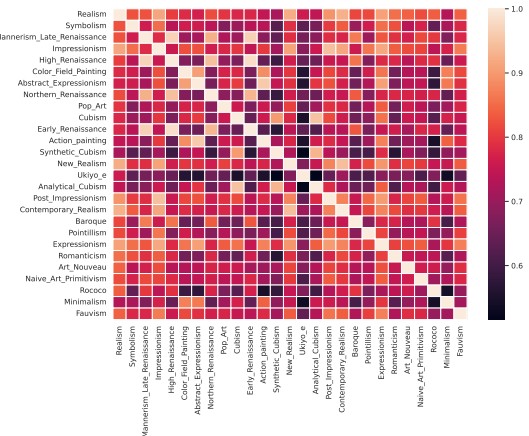

Figure B.5: All pair cosine similarities of generated prompts from Wikiart [23] dataset.

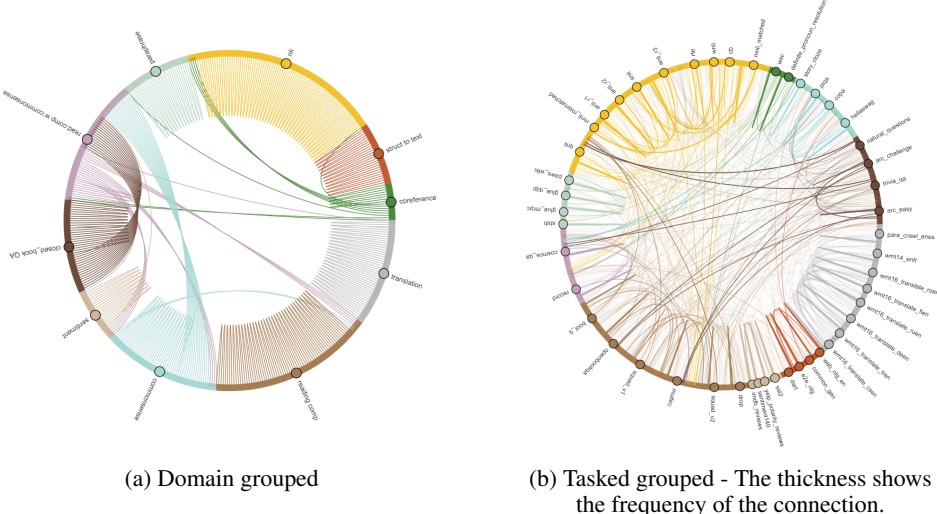

(a) Domain grouped

(b) Tasked grouped - The thickness shows
the frequency of the connection.

Figure C.1: Actual retrieved LoRAs for given domain or task.

match. This is due to its vulnerability to additional retrieved LoRAs, where retrieving the correct LoRA and irrelevant ones can sufficiently lower the exact match score. In general, AC-LoRA consistently retrieves at least one LoRA that belongs to the same domain, though this is not always exclusive for certain domains. As illustrated in Fig. C.1, for queries originating from domains such as CLOSED_BOOK QA and COMMONSENSE, AC-LoRA occasionally retrieves LoRAs from other domains. This is intuitive, given that these domains have less distinct boundaries compared to other tasks. If the primary goal of AC-LoRA were to retrieve LoRAs based on the task, rather than knowledge, performance in these cases could be improved by emphasizing the requested format more during retrieval.

One can notice a similar behavior in Fig. C.2 and Fig. C.3, where we show an extensive version of Fig. 5. The plots show the results with increasing threshold (horizontally) and fetch_k parameter (vertically). The threshold indicates that LoRAs with a retrieved average similarity score lower than the given threshold are disregarded. The threshold does not significantly affect the results, except for thresholds higher than $0.5$, where we start to retrieve fewer, if any, LoRAs, and thus also not the correct one. On the other hand, fetching more documents has a minimal impact on the results. While the plots per task (instead of per domain) are a bit noisier, they show a similar trend. The worst-performing one is mnli_mismatched, which is not surprising, as we have not included it in our database (since the entire idea of this task is to see how it performs out of distribution to the matched ones), and therefore cannot be retrieved. In the event that we consider mnli_matched as the correct LoRA in this case, we achieve, for example, a 92% accuracy for the mnli_mismatched queries with hyperparameters k = 10, fetch_k = 200, and threshold = 0.0. Similarly, the task arc_easy and arc_challenge, or anli_{r1,r2,r3}, whose accuracy increases when considering any of the options as correct.

## C.2 RepLiQA

In Fig. C.4 we present a more detailed version of the plot presented in Fig. 5. The plots show the results with increasing threshold (horizontally) and increasing fetch_k parameter (vertically). Similarly to Flan, one can see that the fetch_k parameter does not appear to affect the results significantly. At the same time, once we increase the threshold to $0.5$, retrieval results degrade significantly as this leads AC-LoRA to disregard often all retrieved LoRA. We show the actual retrieved LoRAs per domain and their frequency in Fig. C.5.

### C.2.1 Hinting

To evaluate the Hinting mechanism described in Sec. 3 we run AC-LoRA once by masking (not permitting) the LoRA of the corresponding topic and once with all permissions if there has been a

Table C.1: Full comparison with LoRARetriver.

| Task/Llama27b | Perfect Selection | Selection | | Fusion | | Mixture | | MoE Top1 | MoE Top3 | MoE Soft | SME-AR | Adapter Soup | LoRa Hub | AC-LoRA (95%-CI) (k=3, fetch_k=10) |
|---|---|---|---|---|---|---|---|---|---|---|---|---|---|---|
| | | IID | OOD | IID | OOD | IID | OOD | | | | | | | |
| **Struct to Text** | | | | | | | | | | | | | | |
| WebNLG$^{Rouge-1}$ | 71.2 | 67.0 | 53.9 | 49.4 | 45.4 | 57.8 | 53.9 | 45.1 | 47.6 | 49.1 | 51.1 | 3.9 | 32.5 | **69.8** (65.7-75.2) |
| WebNLG$^{Rouge-2}$ | 50.6 | 44.5 | 30.0 | 25.9 | 24.1 | 33.5 | 29.4 | 22.6 | 25.8 | 26.1 | 27.9 | 0.9 | 17.3 | **48.4** (42.2-55.8) |
| WebNLG$^{Rouge-1}$ | 64.4 | 60.9 | 49.1 | 45.5 | 41.0 | 52.3 | 49.6 | 40.0 | 41.9 | 43.3 | 45.4 | 3.9 | 31.1 | **61.9** (56.8-67.6) |
| DART$^{Rouge-1}$ | 71.7 | 67.9 | 58.4 | 56.3 | 53.4 | 63.2 | 60.0 | 55.4 | 56.3 | 56.9 | 60.0 | 3.3 | 40.0 | **72.5** (66.6-76.1) |
| DART$^{Rouge-2}$ | 49.1 | 45.8 | 34.9 | 32.3 | 30.6 | 36.6 | 35.4 | 30.3 | 31.0 | 30.8 | 33.0 | 1.3 | 20.1 | **49.1** (42.4-55.0) |
| DART$^{Rouge-l}$ | 64.6 | 61.1 | 52.4 | 50.3 | 47.9 | 56.3 | 52.4 | 49.7 | 50.8 | 50.2 | 54.8 | 3.3 | 35.2 | **64.0** (58.7-69.6) |
| E2ENLG$^{Rouge-1}$ | 66.1 | 65.8 | 59.3 | 62.2 | 57.2 | 66.0 | 58.7 | 52.9 | 54.0 | 55.3 | 53.2 | 4.2 | 50.1 | **66.1** (62.0-70.3) |
| E2ENLG$^{Rouge-2}$ | 40.0 | 39.4 | 34.1 | 34.7 | 32.0 | 38.8 | 32.1 | 26.9 | 27.6 | 28.8 | 27.5 | 2.4 | 26.3 | **39.6** (35.6-43.5) |
| E2ENLG$^{Rouge-1}$ | 56.7 | 55.7 | 50.2 | 52.7 | 49.1 | 56.9 | 49.0 | 45.1 | 45.0 | 45.1 | 42.2 | 4.2 | 42.2 | **56.4** (52.5-60.6) |
| CommonGen$^{Rouge-1}$ | 46.9 | 44.7 | 29.0 | 29.9 | 27.7 | 36.5 | 29.0 | 29.0 | 29.3 | 30.1 | 27.6 | 6.6 | 19.8 | 38.3 (30.9-44.0) |
| CommonGen$^{Rouge-2}$ | 18.8 | 18.3 | 7.3 | 9.9 | 7.2 | 11.1 | 8.6 | 7.7 | 7.1 | 9.3 | 8.4 | 0.0 | 6.9 | 11.1 (6.7-16.2) |
| CommonGen$^{Rouge-1}$ | 42.5 | 40.5 | 24.0 | 25.8 | 23.3 | 32.7 | 24.8 | 24.4 | 25.1 | 26.3 | 24.3 | 6.6 | 18.0 | 34.8 (28.2, 40.4) |
| **Translation** | | | | | | | | | | | | | | |
| Paracrawl-enes | 24.3 | 24.2 | 20.3 | 22.9 | 22.3 | 22.8 | 22.1 | 18.0 | 18.8 | 19.5 | 21.6 | 4.5 | 16.4 | **26.3** (19.0-33.6) |
| WMT'16-tren | 3.2 | 3.1 | 2.6 | 3.5 | 3.3 | **3.7** | 2.6 | 3.5 | 3.2 | 3.4 | 3.2 | 0.0 | 2.0 | 3.4 (0.2-8.3) |
| WMT'16-ruen | 10.8 | 10.4 | 9.8 | 9.2 | 9.3 | 11.0 | 10.8 | 6.2 | 7.8 | 8.3 | 7.3 | 0.0 | 4.8 | **11.3** (6.0-17.2) |
| WMT'16-deen | 18.9 | 18.7 | **20.3** | 17.9 | 18.8 | 18.8 | 18.7 | 11.6 | 14.0 | 14.7 | 16.6 | 1.1 | 11.4 | 17.9 (12.1-24.5) |
| WMT'16-fien | 6.5 | 6.5 | 7.0 | 7.2 | 7.1 | 7.3 | **7.8** | 6.2 | 6.2 | 6.1 | 6.5 | 0.7 | 4.3 | 7.7 (2.6-12.9) |
| WMT'16-roen | 13.9 | 14.0 | 12.3 | 12.8 | 13.3 | 13.1 | 12.2 | 9.8 | 10.7 | 10.1 | 10.3 | 0.3 | 8.0 | **15.1** (9.4-20.7) |
| WMT'14-enfr | 16.5 | 16.1 | 16.9 | 17.7 | 18.0 | 17.8 | 18.0 | 15.9 | 17.3 | 17.1 | 16.4 | 3.5 | 15.2 | 17.9 (12.2-21.9) |
| WMT'16-csen | 10.7 | 9.4 | 7.0 | 6.1 | 6.2 | 8.3 | 5.8 | 4.7 | 6.3 | 6.3 | 6.3 | 0.8 | 6.1 | **9.7** (5.4-13.8) |
| **Commonsense** | | | | | | | | | | | | | | |
| StoryCloze | 72.0 | 62.0 | 42.0 | 72.0 | 68.0 | 84.0 | 58.0 | 74.0 | 70.0 | 70.0 | 68.0 | 62.0 | 48.0 | **86.0** (76.0-96.0) |
| PIQA | 46.0 | 46.0 | 32.0 | 34.0 | 36.0 | 38.0 | 34.0 | 40.0 | 38.0 | 38.0 | 36.0 | 38.0 | 0.0 | 44.0 (31.9-58.0) |
| COPA | 86.0 | 74.0 | 68.0 | 78.0 | 70.0 | 80.0 | 68.0 | 72.0 | 70.0 | 72.0 | 70.0 | 56.0 | 22.0 | **80.0** (72.0-92.0) |
| HellaSwag | 46.0 | 40.0 | 42.0 | 20.0 | 18.0 | 44.0 | 40.0 | 32.0 | 30.0 | 26.0 | 26.0 | 28.0 | 0.0 | **50.0** (36.0-64.0) |
| **Sentiment** | | | | | | | | | | | | | | |
| SST-2 | 98.0 | 98.0 | 96.0 | 74.0 | 78.0 | 96.0 | 94.0 | 56.0 | 68.0 | 66.0 | 66.0 | 74.0 | 0.0 | **98.0** (94.0-100.0) |
| Yelp | 98.0 | 94.0 | 94.0 | 96.0 | 96.0 | 98.0 | 98.0 | 86.0 | 90.0 | 86.0 | 84.0 | 80.0 | 0.0 | **98.0** (90.0-100.0) |
| IMDB | 96.0 | 96.0 | 96.0 | 92.0 | 82.0 | 96.0 | 96.0 | 76.0 | 80.0 | 80.0 | 84.0 | 80.0 | 0.0 | **96.0** (90.0-100.0) |
| sentiment140 | 68.0 | 70.0 | 70.0 | 54.0 | 58.0 | 68.0 | **74.0** | 62.0 | 62.0 | 66.0 | 62.0 | 60.0 | 2.0 | 68.0 (56.0-80.0) |
| **READING Comp.** | | | | | | | | | | | | | | |
| MultiRC | 68.0 | 52.0 | 38.0 | 44.0 | 44.0 | 48.0 | 44.0 | 54.0 | 52.0 | 50.0 | 48.0 | 40.0 | 6.0 | **60.0** (46.0-72.0) |
| SQuADv2 | 62.0 | 56.0 | 12.0 | 30.0 | 20.0 | 22.0 | 16.0 | 24.0 | 24.0 | 26.0 | 22.0 | 16.0 | 0.0 | 34.0 (24.0-50.0) |
| SQuADv1 | 68.0 | 66.0 | 68.0 | 64.0 | 64.0 | 62.0 | 68.0 | 68.0 | 70.0 | 66.0 | 66.0 | 54.0 | 4.0 | 56.0 (42.0-70.0) |
| OBQA | 82.0 | 68.0 | 58.0 | 64.0 | 60.0 | 78.0 | 66.0 | 62.0 | 64.0 | 66.0 | 60.0 | 40.0 | 0.0 | 70.0 (56.0-80.0) |
| BoolQ | 84.0 | 60.0 | 60.0 | 68.0 | 70.0 | 80.0 | 76.0 | 74.0 | 68.0 | 76.0 | 70.0 | 72.0 | 6.0 | **84.0** (74.0-94.0) |
| drop | 40.0 | 8.0 | 6.0 | 14.0 | 12.0 | 18.0 | 14.0 | 10.0 | 8.0 | 8.0 | 8.0 | 22.0 | 0.0 | **28.0** (14.0-38.0) |
| **CLOSED-BOOK QA** | | | | | | | | | | | | | | |
| NQ | 18.0 | 16.0 | 10.0 | 16.0 | 14.0 | 16.0 | 10.0 | 12.0 | 12.0 | 12.0 | 4.0 | 12.0 | 0.0 | 10.0 (2.0-18.0) |
| ARC-e | 50.0 | 56.0 | 70.0 | 54.0 | 56.0 | 66.0 | 82.0 | 58.0 | 58.0 | 60.0 | 58.0 | 48.0 | 0.0 | 64.0 (46.0-74.0) |
| ARC-c | 46.0 | 42.0 | 46.0 | 34.0 | 34.0 | 50.0 | 46.0 | 46.0 | 42.0 | 42.0 | 42.0 | 24.0 | 0.0 | 38.0 (24.0-48.0) |
| TriviaQa | 66.0 | 46.0 | 46.0 | 60.0 | 46.0 | 48.0 | 56.0 | 46.0 | 42.0 | 46.0 | 24.0 | 42.0 | 4.0 | 44.0 (24.0-54.0) |
| **COREFERENCE** | | | | | | | | | | | | | | |
| DPR | 54.0 | 50.0 | 50.0 | 56.0 | 60.0 | 68.0 | 56.0 | 64.0 | 60.0 | 62.0 | 62.0 | 46.0 | 2.0 | 54.0 (40.0-66.0) |
| WSC | 50.0 | 50.0 | 42.0 | 38.0 | 46.0 | 58.0 | 42.0 | 58.0 | 58.0 | 52.0 | 54.0 | 40.0 | 0.0 | 54.0 (40.0-68.0) |
| **READ. COMP. W/ COMMONSENSE** | | | | | | | | | | | | | | |
| CosmosQa | 68.0 | 68.0 | 34.0 | 46.0 | 32.0 | 50.0 | 46.0 | 44.0 | 46.0 | 44.0 | 38.0 | 14.0 | 6.0 | **72.0** (58.0-82.0) |
| record | 70.0 | 70.0 | 26.0 | 24.0 | 6.0 | 42.0 | 34.0 | 18.0 | 12.0 | 14.0 | 8.0 | 14.0 | 0.0 | 54.0 (32.0-62.0) |
| **PARAPHRASE** | | | | | | | | | | | | | | |
| Paws Wiki | 90.0 | 64.0 | 40.0 | 44.0 | 42.0 | 56.0 | 46.0 | 56.0 | 50.0 | 48.0 | 54.0 | 60.0 | 2.0 | **78.0** (66.0-88.0) |
| QQP | 74.0 | 74.0 | 68.0 | 66.0 | 60.0 | 80.0 | 58.0 | 50.0 | 40.0 | 36.0 | 28.0 | 54.0 | 0.0 | 74.0 (62.0-86.0) |
| MRPC | 60.0 | 58.0 | 58.0 | 60.0 | 62.0 | 60.0 | 58.0 | 42.0 | 44.0 | 40.0 | 42.0 | 60.0 | 2.0 | 60.0 (44.0-74.0) |
| STSB | 38.0 | 36.0 | 16.0 | 12.0 | 12.0 | 30.0 | 20.0 | 20.0 | 20.0 | 20.0 | 14.0 | 12.0 | 0.0 | **40.0** (26.0-54.0) |
| **NLI** | | | | | | | | | | | | | | |
| CB | 88.9 | 80.0 | 62.2 | 77.8 | 57.8 | 86.7 | 66.7 | 68.9 | 64.4 | 68.9 | 62.2 | 55.6 | 13.3 | 77.8 (62.2-86.6) |
| WNLI | 70.0 | 68.0 | 46.0 | 44.0 | 50.0 | 60.0 | 54.0 | 56.0 | 56.0 | 42.0 | 44.0 | 52.0 | 0.0 | 62.0 (50.0-78.0) |
| ANLI-r1 | 50.0 | 50.0 | 50.0 | 40.0 | 42.0 | 40.0 | 42.0 | 40.0 | 40.0 | 36.0 | 38.0 | 38.0 | 24.0 | 44.0 (30.0-60.0) |
| ANLI-r2 | 46.0 | 46.0 | 46.0 | 32.0 | 36.0 | 46.0 | 46.0 | 40.0 | 36.0 | 38.0 | 32.0 | 46.0 | 20.0 | 42.0 (26.0-54.0) |
| ANLI-r3 | 46.0 | 42.0 | 38.0 | 38.0 | 40.0 | 44.0 | 50.0 | 28.0 | 32.0 | 34.0 | 38.0 | 40.0 | 24.0 | 46.0 (26.0-54.0) |
| MNLI-m | 88.0 | 84.0 | 88.0 | 62.0 | 66.0 | 80.0 | 88.0 | 48.0 | 54.0 | 50.0 | 56.0 | 76.0 | 0.0 | 78.0 (66.0-92.0) |
| MNLI-mm | 92.0 | 90.0 | 94.0 | 64.0 | 82.0 | 88.0 | 90.0 | 48.0 | 48.0 | 50.0 | 60.0 | 84.0 | 2.0 | 90.0 (84.0-98.0) |
| SNLI | 96.0 | 84.0 | 84.0 | 56.0 | 58.0 | 90.0 | 92.0 | 54.0 | 52.0 | 54.0 | 54.0 | 82.0 | 0.0 | **96.0** (90.0-100.0) |
| QNLI | 94.0 | 94.0 | 26.0 | 46.0 | 48.0 | 74.0 | 38.0 | 56.0 | 56.0 | 54.0 | 60.0 | 70.0 | 0.0 | 78.0 (64.0-88.0) |
| RTE | 52.0 | 62.0 | 72.0 | 54.0 | 58.0 | 70.0 | 76.0 | 64.0 | 58.0 | 56.0 | 64.0 | **80.0** | 22.0 | 74.0 (62.0-86.0) |

Table C.2: Mapping index for Figs. 3(b) and 6(b) to RepLiQA domain.

| Index | 1 | 2 | 3 | 4 | 5 | 6 | 7 | 8 |
|---|---|---|---|---|---|---|---|---|
| Domain | Regional Folklore and Myths | Local Health and Wellness | Local Environmental Issues | Neighborhood Stories | Local Sports and Activities | Local Technology and Innovation | Local Arts and Culture | Cybersecurity News |

| Index | 9 | 10 | 11 | 12 | 13 | 14 | 15 | 16 | 17 |
|---|---|---|---|---|---|---|---|---|---|
| Domain | Local Politics and Governance | Small and Medium Enterprises | Local News | Local Economy and Market | Local Education Systems | News Stories | Company Policies | Regional Cuisine and Recipes | Incident Report |

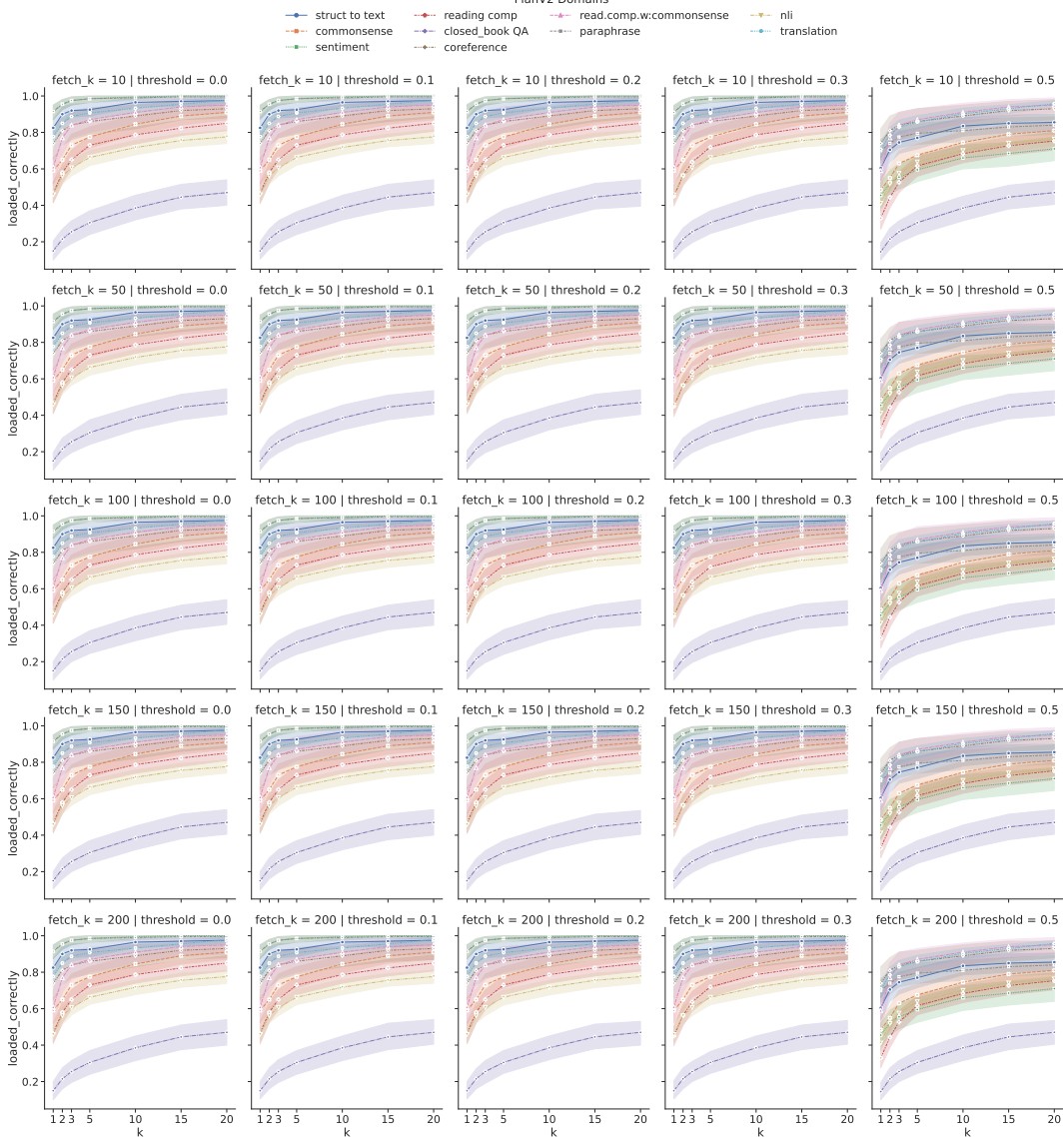

Figure C.2: Flan-V2 task retrieval grouped per domain.

hint in the masked run. In Fig. C.6 we present the results. As one can see, on average, the hinted LoRA can improve the answer given by AC-LORA.

### C.2.2  Combining Knowledge

QUESTION: What are two significant contributions Dr. Chase has made to AI-powered cybersecurity?

REFERENCE ANSWER: On October 15, 2023, Dr. Chase presented her groundbreaking work on AI-powered cybersecurity at the Chicago Cyber Security Summit. Additionally, in November 2023, her AI-powered security system detected an attempted zero-day vulnerability exploit against a financial institution in Chicago before it could cause significant damage.

LORA1: Dr. Chase has developed an AI-powered cybersecurity platform that uses machine learning algorithms to detect and respond to cyber threats in real-time. Additionally, she has introduced a new cybersecurity curriculum at the university level, which includes topics such as AI ethics and security protocols.

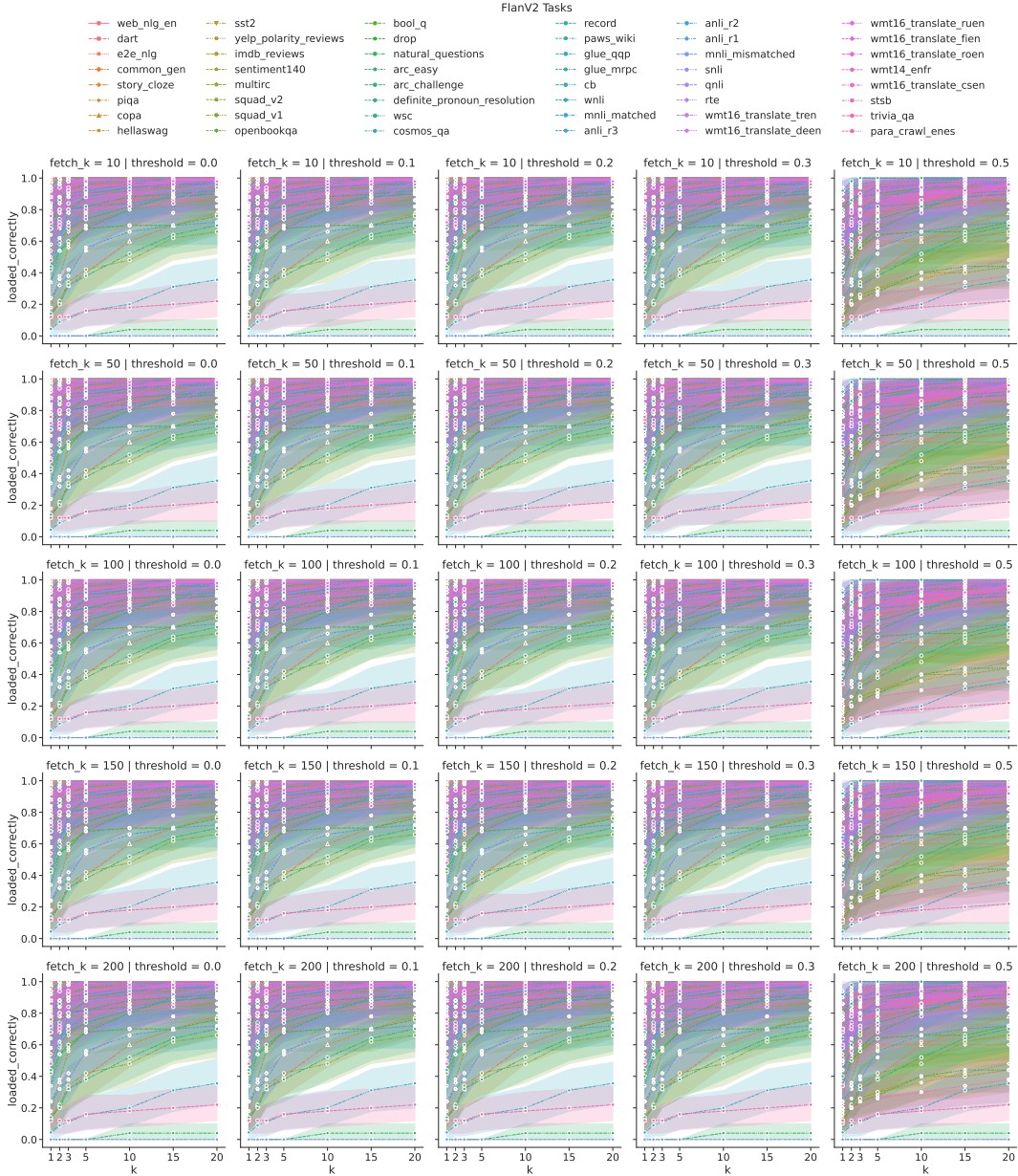

Figure C.3: Flan-V2 task retrieval.

> LoRA2: Dr. Chase has contributed to the development of AI-powered cybersecurity by creating a framework that integrates machine learning and human expertise to detect and respond to cyber threats. Additionally, she has demonstrated the effectiveness of AI in detecting and mitigating cyber attacks, showcasing its potential to revolutionize the field.
> LoRA12: Dr. Chase has developed an AI system capable of detecting and neutralizing cyber threats in real-time, and she has also created an AI-powered cybersecurity platform that uses machine learning to predict and prevent cyber attacks.

## C.3 Mixing all LoRAs

The primary advantage of using only the relevant LoRAs instead of just mixing all permissible ones is the reduction of inference latency. However, we also observed an average grade improvement of 0.066 (95% interval: (-0.00748, 0.138)) over mixing all possible LoRAS for the RepliQA dataset.

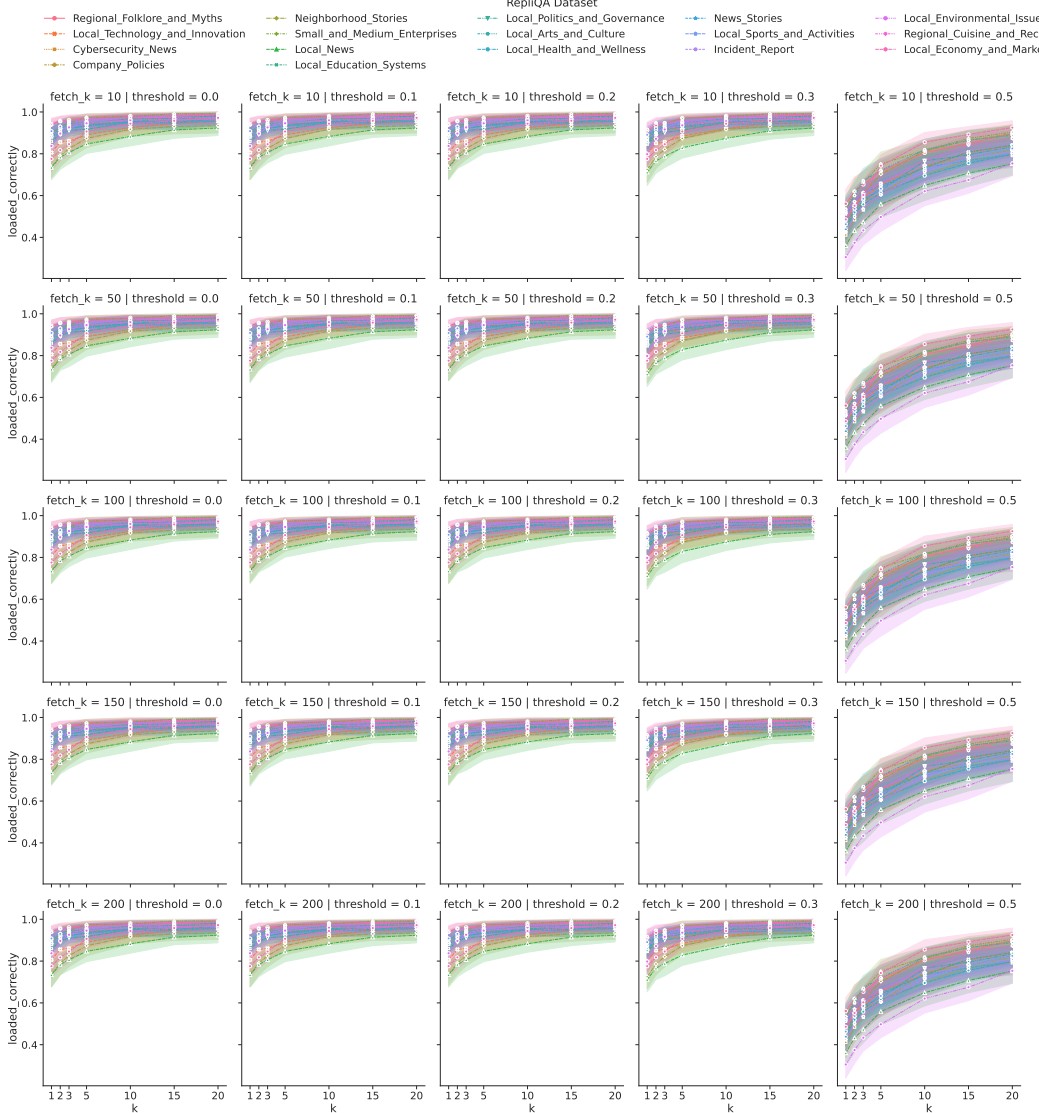

Figure C.4: RepliQA retrieval.

This effect could come from several factors; we suspect it's primarily due to the relatively high rank used with a relatively small dataset. We expect that the AC-LoRA improvement will increase further as the number of LoRAs increases. Flanv2, however, behaves differently. Since Flanv2 is primarily finetuned to influence output format, mixing all Flan LoRAs significantly degrades performance. This is evident in Tab. C.3, which shows the improvement of AC-LORA over the mixing of all Flan LoRAs.

## C.4 Multi-Modal

In the following we present some additional information about our setup for the stable diffusion experiments and show some additional results. We then briefly provide some information regarding the capabilities to use AC-LORA also with other modalities, such as text-image to text.

In general, we view the results presented in this section as more of a proof of concept rather than a comprehensive evaluation. A more thorough analysis would require significantly more resources to accurately assess the capabilities of the base model and the specific contributions made by the finetuning. To ensure a fair and precise evaluation, one would need to create a new dataset (to guarantee that the base model has not previously been trained on it). However, even with this step,

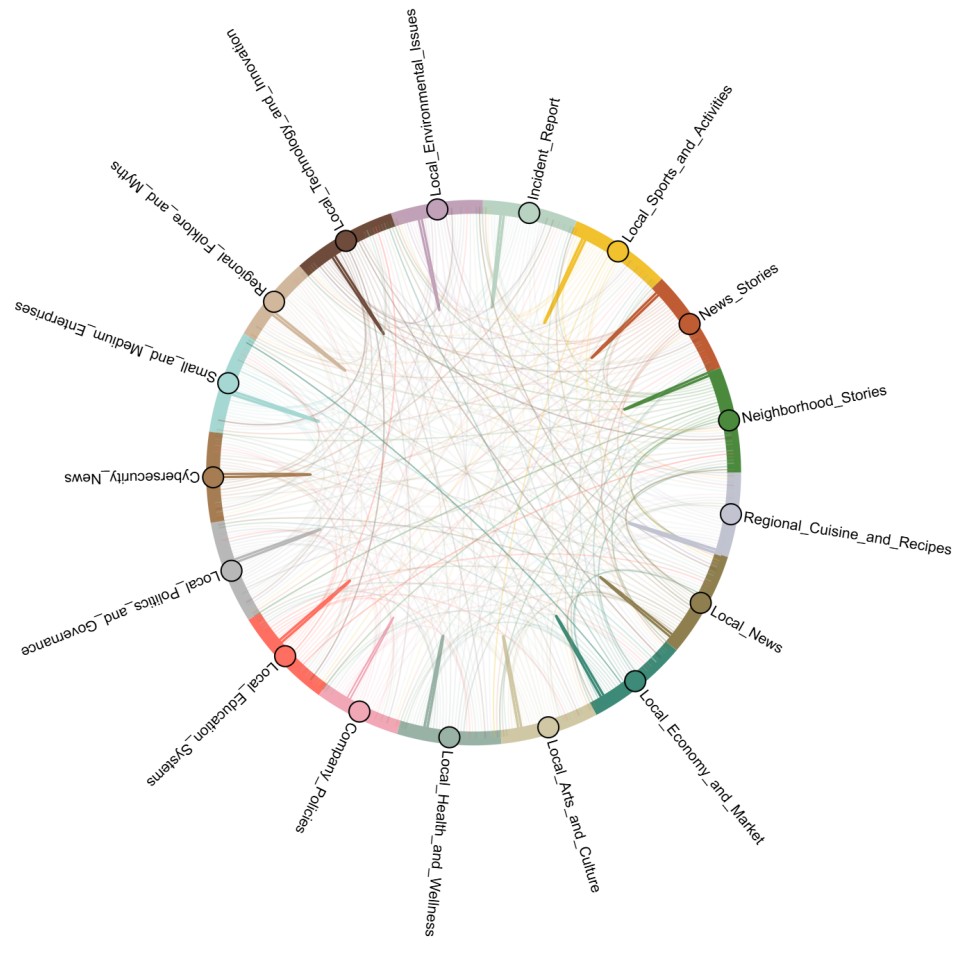

Figure C.5: Actual retrieved LoRAs for given domain. The thickness shows the frequency of the connection.

Table C.3: Improvement of AC-LoRA over just mixing all FlanV2 LoRAs

| Task | Improvement |
|------|-------------|
| Struct to text (rouge-1) | 34.64 (31.95, 37.19) |
| Struct to text (rouge-2) | 26.94 (23.97, 30.17) |
| Struct to text (rouge-l) | 30.64 (28.02, 33.64) |
| Translation | 9.76 (8.22, 11.43) |
| Commonsense | 65.50 (58.50, 72.50) |
| Sentiment | 90.00 (85.50, 94.00) |
| Reading Comprehension | 54.67 (49.32, 60.00) |
| Closed book QA | 36.50 (30.00, 43.00) |
| Coreference | 54.00 (44.00, 64.00) |
| Read.comp.w/commonsense | 59.00 (49.97, 68.00) |
| Paraphrase | 63.00 (56.00, 69.50) |
| NLI | 68.28 (64.44, 72.32) |

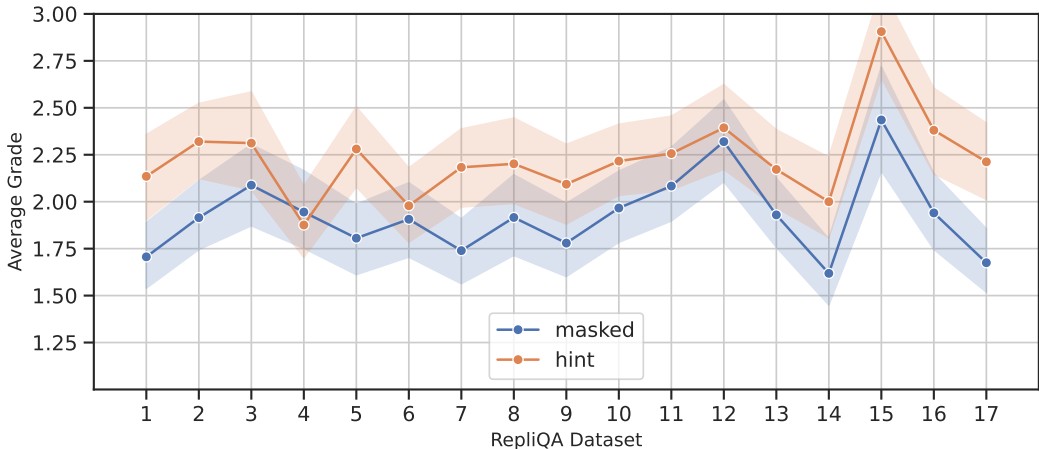

Figure C.6: RepliQA evaluation once with the expected LoRA masked and once with it if part of the Hint.

evaluating the model would remain challenging, as images are inherently more difficult to grade than text. Given these considerations, we believe such an in-depth evaluation is beyond the scope of this work.

### C.4.1 Stable Diffusion

We trained different LoRA models on the different styles in (`WikiArts`[23]). For this, we asked QWEN2-VL to generate a generation prompt given the image, the style, and the artist. From these prompts and the images, we fine-tune STABLE-DIFFUSION-V1-4 on each of the 27 styles, using rank and alpha 16, learning rate 1e-04, and utilizing the diffusers Huggingface library. We then use the generated prompts to build our embeddings for the retriever.

Fig. C.7 shows six example AC-LORA image generation along with their generation prompts and the corresponding retrieved (and mixed) LoRAs.

### C.4.2 Text-Image to Text (Qwen2-VL)

We evaluate AC-LORA also on text-image to text models. We finetune 10 LoRAs using QWEN2-VL-7B-INSTRUCT. Starting from the MMSci dataset [58], we create 10 smaller datasets (5k data points each) as shown in Tab. C.4. We show the retrieval results in Fig. C.8. We describe how we embed the text and image for the retrieval mechanism in Appx. E.

### C.5 Latency

In Sec. 4 we provide our evaluation result of the AC-LORA's time to first token generation with an increasing number of active LoRAs (i.e., isolated permission zones). However, this assumes that the LoRAs and the base model are already loaded into the device's memory (such as the GPU). This is a valid assumption, as switching the model frequently can adversely affect token generation, specifically the latency to first token generation. We evaluate the worst-case scenario, where every user query requires the LoRAs to be loaded into the device memory from scratch. Fig. C.9

## D Templates

### D.1 Knowledge Combination

As described in Sec. 4.1 we built our own dataset starting from RepLiQA split 0 to showcase the capabilities of combining LoRAs to combine knowledge. Starting from all the documents of the CYBERSECURITY NEWS category, we ask deepseek-r1:32b to extract the most relevant facts using the following prompt:

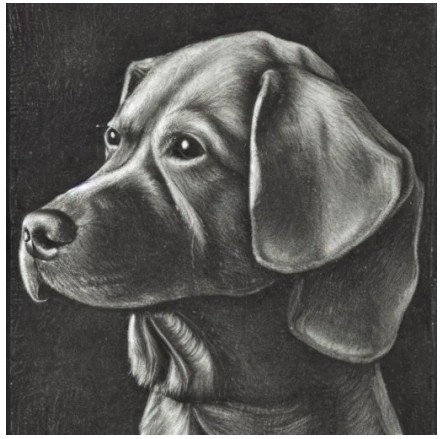

(a) "A dog in style of Da Vinci" - LoRAs: 'Early Renaissance', 'Mannerism Late Renaissance', 'Northern Renaissance'

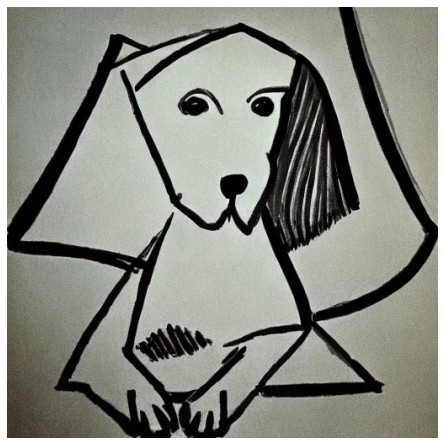

(b) "draw a dog by Picasso" -
LoRAs: 'Symbolism', 'Expressionism', 'Cubism'

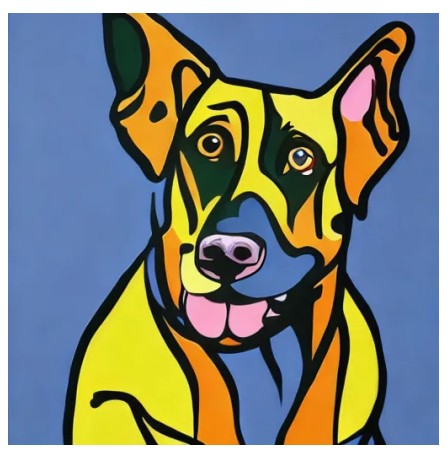

(c) "a dog in pop art style" - LoRAs: 'Pop Art'

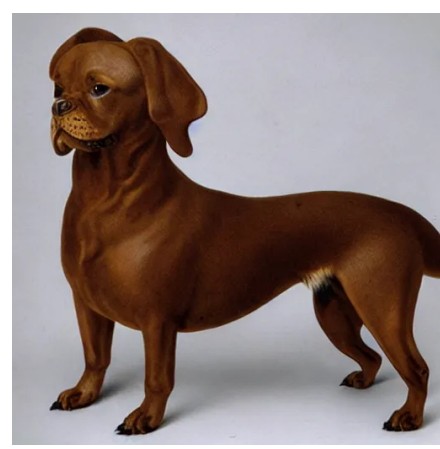

(d) "please generate a rococo dog" - LoRAs: 'Rococo'

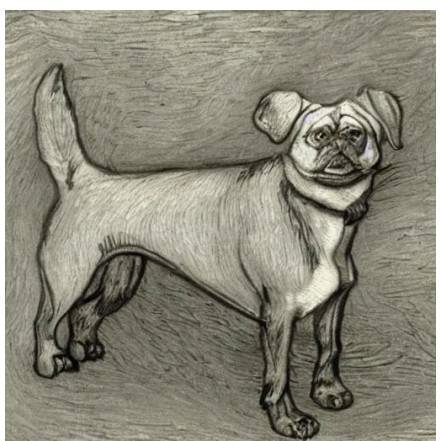

(e) "Please generate an image of a dog as if van gogh would have drawn it" - LoRAs: 'Realism', 'Post Impressionism'

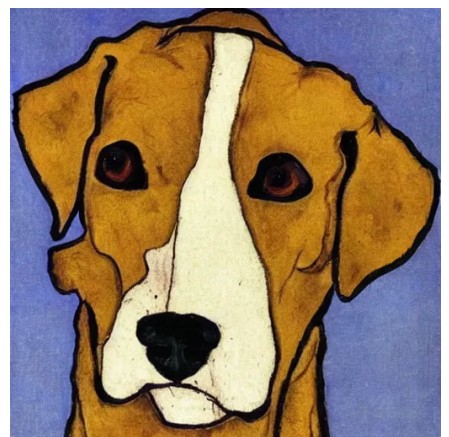

(f) "a dog by Schiele" -
LoRAs: 'Rococo', 'Pop Art',
'Romanticism'

Figure C.7: AC-LoRA STABLE-DIFFUSION-V1-4.

Table C.4: Composition of the different training-sets starting from the MMSci [58] dataset.

| LoRA | Subject | Number of datapoints |
|---|---|---|
| environmental earthscience | Ecology | 3051 |
| | Biogeochemistry | 466 |
| | Hydrology | 119 |
| | Solid Earth sciences | 1022 |
| | Environmental sciences | 342 |
| chemistry chemicalsciences | Biochemistry | 1326 |
| | Chemical biology | 279 |
| | Chemistry | 1249 |
| | Materials science | 2146 |
| engineering technologicalinnovation | Optics and photonics | 645 |
| | Materials science | 2896 |
| | Nanoscience and technology | 1047 |
| | Energy science and technology | 160 |
| | Engineering | 252 |
| neuroscience psychology | Neuroscience | 3400 |
| | Anatomy | 302 |
| | Physiology | 1096 |
| | Neurology | 121 |
| | Psychology | 81 |
| biomedical healthsciences | Microbiology | 1511 |
| | Oncology | 209 |
| | Immunology | 1665 |
| | Diseases | 1240 |
| | Pathogenesis | 375 |
| socialsciences globaldevelopment | Risk factors | 913 |
| | Environmental social sciences | 2127 |
| | Social sciences | 1559 |
| | Business and industry | 156 |
| | Developing world | 245 |
| computational datasciences | Computational biology and bioinformatics | 3295 |
| | Systems biology | 1705 |
| agriculture lifesciences | Ecology | 2184 |
| | Evolution | 1069 |
| | Plant sciences | 1366 |
| | Zoology | 365 |
| | Agriculture | 16 |
| genomics biotechnology | Biochemistry | 1729 |
| | Molecular biology | 1485 |
| | Stem cells | 337 |
| | Genetics | 994 |
| | Biotechnology | 455 |
| space physicalsciences | Physics | 3287 |
| | Space physics | 25 |
| | Optics and photonics | 1216 |
| | Solid Earth sciences | 383 |
| | Astronomy and planetary science | 89 |

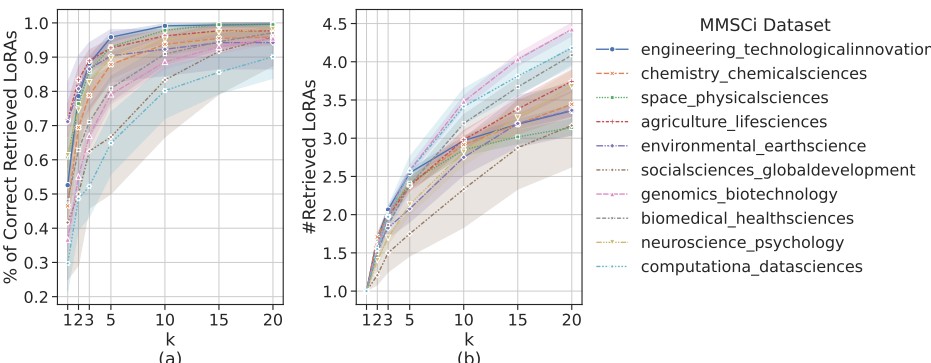

Figure C.8: AC-LORA retrieval results for MMSci based dataset (Tab. C.4) for fetch_k=10 and threshold=0.0

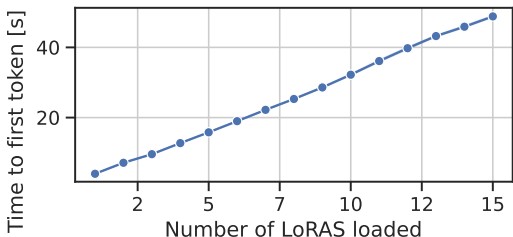

Figure C.9: Time to first token generation latency for a 64-token input query where are all the LoRAs are loaded from scratch to device memory.

> SYSTEM: You are an expert analyzer. Given a text, extract the main (distinct) facts in a concise manner as a list, separated by '\n*'. Each fact must be fully self-contained, meaning it should make complete sense on its own without requiring any context from the original text or other extracted facts. \n* Always explicitly state the subject and object—never use pronouns (e.g., he, she, they, it) when a clear noun can be used instead. \n* Do not assume or infer any information that is not explicitly stated in the text. \n* Each fact must stand alone—no fact should depend on a previous one to be understood. \n* Keep facts as concise, accurate, and clear as possible while maintaining completeness. There should always be an even number of facts (between 2 and 10).
> USER: {extracted document}

We then divide randomly the generated facts into two groups LORA-1 and LORA-2. From these, we generate one new article for each using the following prompt:

> SYSTEM: You are an expert writer. Given a list of facts, write a coherent and well-structured text that includes only the provided facts—nothing more, nothing less. Ensure the text is readable, logically structured, and flows naturally while maintaining clarity and conciseness. Do not add any additional information or interpretation beyond the given facts.
> USER: List of facts: fact_1 \n ... \n fact_n

From this we now generate two different types of question and answer pairs.
**Single-LoRA QA:** These questions and answers should be answerable only by one LoRA. We take two facts from the same set (either LORA-1 or LORA-2) and their corresponding generated articles, and ask the model to generate a question and answer pair that is only answerable when knowing both facts. To do this, we use the following prompt:

> SYSTEM: You are an expert question generator. Given two facts and a context, create a question and answer pair where the answer requires both facts to be answerable.\n \n - Clearly name the subject and object in both the question and answer.\n- Do not infer any information that is not explicitly stated in the facts or the Context.\n- Do not add any additional explanation—only

> provide the question and answer.
> USER: context: {context_lora_n}\n fact_1: {fact_1_lora_n}\n fact_2:{fact_2_lora_n}

where n is either 1 or 2.

**Combined-LoRA QA:** In this case, we take one fact from each set (one from LORA-1 and one from LORA-2) and both generated articles and ask the model to generate a question and answer pair which is only answerable when knowing both facts. To do this, we use the following prompt:

> SYSTEM: You are an expert question generator. Given two facts, and two contexts create a question and answer pair where the answer requires both facts to be answerable.\n \n- Clearly name the subject and object in both the question and answer.\n- Do not infer any information that is not explicitly stated in the facts or the Contexts.\n- Do not add any additional explanation - only provide the question and answer.
> USER:      context_1:{context_lora_1}\n   context_2:{context_lora_2}\n   fact_1:{fact_lora_1}\n fact_2:{fact_lora_2}"

We then review the cases in which the question and answer pair were not in the correct format, which occurred in very few cases ($< 10$). Afterwards, we create two training sets (one for each LoRA) and one test set. In each training set, we include:

- 250 single-LoRA questions of the corresponding set, once with and once without context (i.e., 500 data points). We also ensure here that each context generated appears at least once in this set.
- 140 single-LoRA questions of the corresponding set without context.

So, in total, each of the training sets contains 640 data points. The test set comprises all the combined LoRa questions and the remaining single-LoRa questions (a total of 1,065 data points).

We use Prompt 9 to grade our evaluation.

## D.2 WikiArts

We use QWEN2-VL-7B-INSTRUCT to generate two-generation prompts for each image in the WikiArts dataset [23]. For this, we input the image and the following prompt:

> Given the style, a genre, the artist which we try to reproduce and an image please write **two** generation prompt for the given image. It should be one or two sentences per prompt. Do *only* write the prompts, separate them always only by a new line ('\n').\n Style:{style}, Genre:{genre}, Artist:{artist}

We use these prompts for both finetuning the model and for building the vector database for later retrieving the correct LoRA.

For the images displayed in Fig. 7 we use the following prompt:

> "a serene Buddhist temple on a mountain path, captured in peaceful brushwork"

## D.3 Grading

### D.3.1 Flan

We use the same grading functions from Zhao et al. [44] to evaluate our results, to ensure comparability. Therefore, we evaluated it using the BLEU score from the Natural Language Toolkit [59] and the Rouge score from the Rouge Python package.

### D.3.2 RepLiQA

To evaluate the different experiments on the RepLiQA dataset, we use GEMMA-3-27B to give each generated answer a grade between 1 and 5.

The prompt we use is the following:

Table E.1: Hyperparameters used to finetune RepLiQA LoRAs

| Hyperparameter | Values |
|---|---|
| base model | meta-llama/Llama-3.1-8B-Instruct |
| epochs | 3 |
| per_device_train_batch_size | 4 |
| gradient_accumulation_steps | 8 |
| learning_rate | 1e-4 |
| lora_alpha | 64 |
| r | 64 |
| lora modules | o_proj, k_proj, gate_proj, down_proj, v_proj, q_proj, up_proj |

Table E.2: Number of datapoints used for building the vector base for different tasks.

| Task | anli_r1 | cb | rte | mnli matched | wnli | dpr | wsc | copa | story cloze | glue mrpc | arc challenge | arc easy | openbook qa | *All other Tasks* |
|---|---|---|---|---|---|---|---|---|---|---|---|---|---|---|
| **# Datapoints** | 15k | 500 | 8k | 3k | 1.9k | 3.8k | 1.6k | 1.7k | 5.5k | 12k | 3k | 7.2k | 15k | 30k |

> SYSTEM: Evaluate how well the Generated Answer matches the Reference Answer or the detailed reference answer for the given Query. Be strict: Names, dates, and specific details must be exact to be correct. Additional facts that are not in the Reference Answer do not affect the score unless they contradict the Reference Answer, in which case the score should decrease. If a name, date, or key fact is incorrect, the score must be 1, regardless of other details. Assign a score from 1 to 5 based on accuracy, completeness, and relevance: 5 = Identical meaning, all details correct. 4 = Mostly correct, with only minor wording variations but the same meaning. 3 = Partially correct, with some missing or incorrect details. 2 = Weak relevance, with significant errors or omissions. 1 = Incorrect or unrelated. Input Format: Query: query \n Reference Answer: reference_answer \n Generated Answer: generated_answer \n Output Format: Explanation: [Brief reason for the score] Score: [1-5]
> USER: Query: {query}\n Reference Answer: {reference_answer} \n Generated Answer: {generated_answer}

In case we have two reference answers, for example, for most of our RepLiQA experiments, we also add the long reference answer in addition to the reference answer as 'detailed reference answer' to the prompt.

# E   Implementation detail

## E.1   Evaluation Setup

We run our experiments on two workstation GPUs, each with 10752 processing cores, 48GB GDDR6 VRAM. (384-bit bus and 768 GB/s memory bandwidth), and a 38.7 TFLOPS single precision performance. The GPU is connected to a host ($2\times$ x86 44-core CPU with 256 GB RAM) over a PCIe 4.0.

## E.2   Finetuning

### E.2.1   RepLiQA

For language models, we use unsloth [60] to fine-tune the different LoRAs as the library is faster and saves memory compared to the base implementation. We finetune the 17 LoRAs for the RepLiQA dataset with the hyperparameters displayed in Tab. E.1.

For the knowledge injection experiments displayed in Fig. 3(a) for cybersecurity, we maintain the same hyperparameters and only adjust the values for alpha and rank. Also, we fine-tune it for 10 epochs and save the LoRA at every epoch to study overfitting.

### E.2.2 FlanV2

As mentioned before, we did not fine-tune the FlanV2 LoRAs as we use the one made available by the authors from [44]. As they only focused on formats, they finetuned their LoRAs by only targeting the v_proj and q_proj modules.

### E.2.3 WikiArts

To finetune the different WikiArts LoRAs, we use the Huggingface diffusers library [61]. We set rank and alpha both to 16.

### E.2.4 Retriever

We utilize the LangChain [62] library to implement most of our retrieval process, and its FAISS [63] implementation serves as our vector store.

### E.2.5 Building the database

**Text.**
As the training set for FlanV2 used for the different LoRAs was not shared, we constructed one based on the official FlanV2 dataset for the retriever. In particular, we take the first 30k (or fewer for smaller tasks) samples of each selected task as the training set. The exact number of datapoints per task is shown in Tab. E.2 For WikiArts, we generate prompts for the different images in the training set and use these for the database. For RepLiQA, we used a first version of the training set, with only two entries per data point: one with and one without context.

We then use SENTENCETRANSFORMERSTOKENTEXTSPLITTER and the embedding model ALL-MNET-BASE-V2 [50] to split the files into chunks of 100 tokens and create the FAISS vector store by adding the created documents.

**Text-Image.**
We embed each text and image together using the multi-modal embedding model INFGRAD/-JASPER_EN_VISION_LANGUAGE_V1 [64]. We initiate a SENTENCETRANSFORMER with this model.

### E.2.6 Retrieving and Hinting

We use the *similarity_search_with_score_by_vector* function to retrieve the most likely LoRAs for text-image inputs and *similarity_search_with_score* for only-text queries.

We use the filter function to enforce access control, retrieving only the embeddings with the allowed LoRAs in the metadata.

The Hinting mechanism is implemented as two database queries, once with the filter function and once without.

### E.3 LoRA Mixing

We patch the PEFT library to enable the mixing. We mainly modified the forward function for the Linear LoRA layers.

## F   List of Assets

The following is a list of assets, along with their licenses (and sources, linked) that we use in this paper.

- RepliQA dataset: CC BY 4.0
- Flan V2 dataset: Apache License Version 2.0, January 2004
- Wikiart dataset: BSD 3-Clause License
- MMSci dataset: CC BY 4.0

- Meta Llama: META LLAMA 3 COMMUNITY
- Google Gemma: Open source
- Qwen models: royalty-free limited license
- all-mpnet-base-v2: Apache License Version 2.0, January 2004
- langchain: MIT
- PEFT: Apache License Version 2.0, January 2004
- Stable-diffusion: CreativeML Open RAIL-M, August 22, 2022

