# OpenReview forum: "AC-LoRA: (Almost) Training-Free Access Control Aware Multi-Modal LLMs"
_NeurIPS.cc/2025/Conference — NeurIPS 2025 poster_

### Official Review · Reviewer_PYQb · 2025-06-04

**Clarity:** 2
**Significance:** 1
**Originality:** 1
**Rating:** 4
**Confidence:** 3

**Summary:**

The paper introduces AC-LoRA, a system that regulates which document(s) an LLM can use (include as finetuned  LoRAs) for users with different roles. AC-LoRA stores a set of LoRAs with the corresponding documents used to fine-tune these LoRAs in a form of sentence embeddings. At inference time, AC-LoRA considers only the LoRAs/documents that are permissible to the user, and compares the embedding of the user’s query to the available document embeddings to dynamically load the relevant LoRAs. After selecting the relevant LoRAs, the model generates an answer by mixing the retrieved LoRAs.

**Questions:**

### Questions:
* Q1: Table 2: what is meant with IID and OOD?
* Q2: Why not use a baseline with only permissible LoRAs?
* Q3: Is AC-LoRA practical in case of hundreds of (relevant) documents?


### Comments:

L165: “Isolated LoRA Fine-tuning and Knowledge Injection”: Please motivate the experiment better at the very beginning ("LoRAs can reliably inject domain-specific knowledge")
Table 2: The selection baseline seems competitive. Please add this to the discussions (L310-319)

Some parts can be made more consistent (e.g., "knowledge merging" is used in 4.1, and the corresponding paragraph in 4.2 is called "combining knowledge")

L206: I believe $Q$ is not defined before

Fig. 5 caption: “Retrieval of FlanV2 target domains (**left**) and corresponding retrieved domains (**left**).” (one should be "right")

**Ethical Concerns:**

["NO or VERY MINOR ethics concerns only"]

**Final Justification:**

The authors addressed my concerns about the evaluation (baseline with permissible LoRAs, evaluating hinting) in the rebuttal. I'm not familiar with this area of research so it is difficult for me to judge how novel/incremental is their approach in combining knowledge.

**Limitations:**

yes

**Quality:**

2

**Strengths And Weaknesses:**

Strengths:
* The paper targets an important problem with the aim of making the use of LLMs more practical
* The paper considers both text and image modalities (the latter is admittedly not comprehensive (cf. Limitations))

Weaknesses:
* The work is incremental. From my understanding, AC-LoRA adds only a filtering step to select only relevant LoRAs based on the cosine similarity. The mixing is already introduced in other works.
* It is not clear how AC-LoRA compares to a baseline where all the permissible LoRAs for the user are used.
* Hinting is introduced as part of AC-LoRA, but is never evaluated and its benefits are not clear (it only shows the users that there is something they don't see)

---

> ### Author Rebuttal · Authors · 2025-07-29
>
> We thank the reviewer for their time and valuable feedback. In the following, we summarize the main concerns and answer them.
>
> ### **Are there any benefits to hinting?**
>
> We envision that systems like AC-LoRA can enhance the information exchange and utilization of AI agents on corporate datasets. It's often the case that some relevant information belongs to other projects or departments that require additional permission or approval due to strict corporate information security regulations. Therefore, for a user who does not have access to the most relevant LoRA to their query, they will only get an inferior response. The hint serves as a guide for the user in accessing this information. Such a feature is currently absent in corporate AI agents.
> We initially chose not to include this evaluation, as we believe the benefits of this feature are already reflected in Figure 3.b and the retrieval experiments. Additionally, the current datasets are not ideal for showcasing their full potential.
> However, to give a clearer picture, we’ve now evaluated the first 20 datapoints per RepLiQA topic—once with the topic LoRA masked, and once with the hinted LoRA (when available). For k = 3, the results (with 95% confidence interval) are as follows:
>
> | **RepLiQA Topics / Mean Grade** | ** With Masked  Topic LoRa** | **With Hinted  LoRA** | **No Hint Answers** |
> |:---:|:---:|:---:|---|
> | **ALL** | **1.870  (1.752, 1.994)** | **2.18 (2.061, 2.331)** | **57/340** |
> | Local Environmental Issues | 1.55 (1.1, 2.0) | 1.8 (1.3, 2.35) | 4/20 |
> | Local Health and Wellness | 1.65 (1.2, 2.15) | 2.45  (1.85, 3.1) | 2/20 |
> | Cybersecurity News | 2.05  (1.598, 2.6) | 2.25 (1.65, 2.95) | 2/20 |
> | Local Sports and Activities | 1.65 (1.25, 2.05) | 2.25 (1.6, 2.95) | 4/20 |
> | Local Arts and Culture | 1.65 (1.3, 2.05) | 2.25 (1.85, 2.65) | 2/20 |
> | Neighborhood Stories | 1.55 (1.2, 2.0) | 1.4 (1.1, 1.8) | 6/20 |
> | Company Policies | 2.4 (1.85, 2.9) | 3.0 (2.55, 3.45) | 1/20 |
> | Local News | 1.9  (1.45, 2.4) | 1.8 (1.35, 2.301) | 6/20 |
> | Local Education Systems | 2.2  (1.65, 2.75) | 2.25 (1.7, 2.8) | 6/20 |
> | Regional Folklore and Myth | 1.5 (1.15, 1.9) | 1.6  (1.2, 2.0) | 0/20 |
> | Incident Report | 1.75  (1.35, 2.25) | 2.3 (1.75, 2.9) | 1/20 |
> | News Stories | 1.6 (1.2, 2.15) | 2.15 (1.5, 2.85) | 0/20 |
> | Local Technology and Innovation | 2.0 (1.5, 2.5) | 2.1 (1.6, 2.651) | 6/20 |
> | Regional Cuisine and Recipes | 2.45 (1.85, 3.00) | 2.45 (1.9, 3.00) | 2/20 |
> | Local Politics and Governance | 1.7 (1.35, 2.1) | 2.05 (1.6, 2.55) | 5/20 |
> | Local Economy and Market | 2.5 (1.9, 3.1) | 2.7 (2.15, 3.2) | 6/20 |
> | Small and Medium Enterprises | 1.7 (1.3, 2.15) | 2.4 (1.9, 2.9) | 4/20 |
>
> *Action point: We’ll include the evaluation over the full dataset in the appendix.*
>
> ### **Baseline with only permissible LoRAs**
>
> The primary advantage of leading only the relevant LoRA is the reduction of inference latency.
> Additionally, we observed an average grade improvement of 0.066 (95% interval: (-0.00748, 0.138)) over all permissible LoRAs in the RepliQA dataset. This effect could come from several factors; we suspect it’s primarily due to the relatively high rank used with a relatively small dataset. We expect that the AC-LoRA improvement will increase further as the number of LoRAs increases. Flanv2, however, behaves differently. Since Flanv2 is primarily finetuned to influence output format, mixing all Flan LoRAs significantly degrades performance.
>
> This is evident in the table below, which shows the improvement of AC-LORA over the mixing of all Flan LoRAs.
>
> | **FlanV2 / Improvement over Mixing All** | **Metric**  | **Mean** | **Lower Bound** | **Upper Bound** |
> |:----------------------------------------:|-------------|:--------:|:---------------:|:---------------:|
> |              struct to text              |   Rouge-1   |   34.64  |      31.95      |      37.19      |
> |              struct to text              |   Rouge-2   |   26.94  |      23.97      |      30.17      |
> |              struct to text              |   Rouge-l   |   30.64  |      28.02      |      33.64      |
> |                commonsense               | Exact Match |   65.5   |       58.5      |       72.5      |
> |                 sentiment                | Exact Match |   90.0   |       85.5      |       94.0      |
> |               reading comp.              | Exact Match |   54.6   |      49.32      |       60.0      |
> |              closed_book QA              | Exact Match |   36.5   |       30.0      |       43.0      |
> |                coreference               | Exact Match |   54.0   |       44.0      |       64.0      |
> |         read.comp.w: commonsense         | Exact Match |   59.0   |      49.97      |       68.0      |
> |                paraphrase                | Exact Match |   63.0   |      56.00      |       69.5      |
> |                    nli                   | Exact Match |   68.2   |      64.44      |      72.32      |
> |                translation               |     Bleu    |    9.7   |       8.22      |      11.43      |
>
>
> ### **Is AC-LoRA practical when there are hundreds of relevant documents?**
>
> The number of relevant LoRAs does not affect the effectiveness of AC-LoRA. We can summarize the effects into 2 scenarios.
> - *All (100s) relevant documents are from the same permission domain (e.g., one LoRA)* - in that case, we choose to assign the average similarity score of the documents to that specific LoRA.
> - *The documents are spread over many permission domains*. In that case, we assign the average similarity score of the relevant document to each permission zone, corresponding to its respective LoRA.
> In both cases, AC-LoRA only calculates the similarity score for the permission domain if the user has access rights to it.
>
> Therefore, unlike RAG, AC-LoRA does not suffer from lower performance (due to increased context size) when retrieving many relevant documents.
> However, there could be a *rare* scenario where a query from a very high-privilege user (e.g., a CEO) triggers loading many LoRAs together. In such a case, the GPU memory will be a bottleneck. This could be mitigated by using a suitable parallelization strategy, such as pipeline parallelization.
>
> *Action point: We will revise the writing to clarify this further.*
>
>
> ### **Incremental improvement**
>
> While it is correct that LoRA mixing is a well-known topic and covered in multiple previous works (see table 1), we want to point out that mixing LoRAs is *not* the primary contribution of this paper. Existing LoRA mixing works focus on mixing task-based LoRAs and composing new tasks from the mixed LoRAs. However, unlike AC-LoRA, they do not address combining knowledge from different LoRAs. The premise of AC-LoRA depends on whether it is feasible to inject new (and possibly disjoint) knowledge into LoRAs. In existing works, there are limited results regarding the usefulness of LoRAs in injecting knowledge. We show that this is indeed possible. Therefore, by mixing LoRAs, we can effectively create responses from multiple permission domains.
> We agree that AC-LoRA is based on observations and techniques from existing work. However, AC-LoRA’s approach to using *training-free* LoRA mixing for *access control* is unique for *multi-modal models* and has not been covered in the literature. Additionally, AC-LoRA provides a complete *end-to-end proof-of-concept* with open-source models and datasets, demonstrating the practicality of such a system.
>
> The selection baseline in Table 2 shows the results with the top retrieved LoRA by LoRARetriver [1]. This does not include any mixing. It is competitive as the Flanv2 dataset only requires exactly one LoRA per task.
>
> [1] LoraRetriever: Input-Aware LoRA Retrieval and Composition for Mixed Tasks in the Wild, Zhao et al.
>
> *Action point: We will improve the discussion about the baseline in the evaluation section and motivate the experiments in the section “Isolated LoRA Fine-tuning and Knowledge Injection” better.*
>
>
> ### **Small typo and format:**
>
> - **Table 2: What is meant by IID and OOD?**
>
> We use the same terminology from LoRARetriever [1], where "IID" signifies that the method can access any LoRA for every test sample, encompassing the LoRA specific to the sample’s task. "OOD" indicates that for each test sample, they mask the LoRA associated with its specific task during the retrieval phase.
>
> *Action point: We will add this to the table caption.*
>
>
> [1] LoraRetriever: Input-Aware LoRA Retrieval and Composition for Mixed Tasks in the Wild, Ziyu Zhao, Leilei Gan, Guoyin Wang, Wangchunshu Zhou
>
>
> - **Consistent naming**
>
> *Action point: We will revise the paper and ensure that all terminologies are consistent.*
>
> - **Q is not defined before**
>
> Line 206: $\mathcal{Q}$ is the user query.
> > for a query $\mathcal{Q}$ the output for each LoRA in each layer …
>
>  - **Typo Fig 5 caption**
>
> *Action point: Yes, will update accordingly.*
>
> We hope we addressed all your questions and would be happy to continue the discussion to clarify any outstanding concerns.

---

> > ### Comment · Reviewer_PYQb · 2025-08-01
> >
> > I thank the authors for their response. The majority of my concerns have been addressed. I increased my score accordingly.

---

> > > ### Author Response · Authors · 2025-08-01
> > >
> > > Dear reviewer, thank you for your reply. Please let us know if you want more clarification or results.

---

### Official Review · Reviewer_WZ6X · 2025-07-01

**Clarity:** 4
**Significance:** 3
**Originality:** 3
**Rating:** 4
**Confidence:** 3

**Summary:**

This paper presents AC-LoRA, the first end-to-end access control-aware inference serving system to equip LLMs with strong access control management. AC-LoRA control the sensitive information access by fine-tuning different LoRAs on predefined data groups and later uses embedding similarity to compute the LoRA based on user permission and user query. The design of similarity combination based on embedding makes AC-LoRA almost training-free. The author conducts comparison experiments with the previous method on two datasets, and the result demonstrates that AC-LoRA show better or on-par performance than SoTA LoRA mixing techniques, which highlights the effectiveness and efficiency of AC-LoRA.

**Questions:**

See weaknesses 3 and 4.

**Ethical Concerns:**

["NO or VERY MINOR ethics concerns only"]

**Final Justification:**

I think the author has addressed my main points during the rebuttal (If the author takes action to elaborate on the final version, if possible), so I keep my positive rating.

**Limitations:**

Yes

**Paper Formatting Concerns:**

there is no issue on paper formatting.

**Quality:**

3

**Strengths And Weaknesses:**

Strengths:

1. I like the content of the paper a lot. The paper is well motivated and written in an organised format.
2. The design of using embedding similarity to implement access control is novel and light-weight. This design not only reduces the additional training requirement but also ensures high performance compared to the previous method.
3. The experiment is comprehensive and sufficient to illustrate the effectiveness, efficiency and potential application of multimodality. Basically, I have no question about the experiment set-up, as the result is pretty clear to support the author's claim.

Weaknesses:
1. The "gap in prior work" section in the introduction is well motivated, but I think there is a lack of the limitation of RAG to the usage of LoRA or LoRA mixing techniques.
2. It seems that the effectiveness of AC-LoRA is built on the sparse distribution of the knowledge base. I am wondering whether AC-LoRA can achieve the same high performance on embedding space that is not well-separated (maybe math or coding)?
3. What if there is partial overlapping between knowledge groups? Can the author illustrate whether the overlapping will affect the LoRA selection?
4. What if one query requires access to more than 5 knowledge groups, for example, 10 or embedding space is pretty large, will GPU memory become a major bottleneck?

---

> ### Author Rebuttal · Authors · 2025-07-29
>
> We thank the reviewer for their time and valuable feedback. In the following, we summarize the main concerns and answer them.
>
> ### **Could you motivate RAG vs LoRA better?**
> This is not a black-or-white issue, and there are capabilities that RAG can achieve that are harder to achieve with LoRAs and vice versa. However, we will summarize our considerations on why RAG is not the best solution in this setting as follows:
>
>
> - With RAG, access control is a bit more straightforward [1] as one can mark the rows in the database with the access control rules (e.g., which user is permitted to access that specific row)
> - Existing work shows that LoRA scores higher in accuracy compared to retrieval [2].
> - RAG often results in a massive context size due to similarity with a larger number of documents. We show in *Figure 8* that the context size is detrimental to the inference latency and memory requirement.
> - Existing works [3]  also show that RAGs are not effective in multi-hop queries.
>
> In a real deployment, one could also consider adding a lightweight RAG (with only the latest added documents that were not yet used to train the LoRA) to AC-LoRA, which partially mitigates RAG’s problem.
>
> *Action point: We will add these points to our motivation section.*
>
> #### References
>
> [1] Privacy-Aware RAG: Secure and Isolated Knowledge Retrieval, Pengcheng Zhou, Yinglun Feng†, Zhongliang Yang
>
> [2] GPT vs RETRO: Exploring the Intersection of Retrieval and Parameter-Efficient Fine-Tuning, Aleksander Ficek, Jiaqi Zeng, Oleksii Kuchaiev
>
> [3] MultiHop-RAG: Benchmarking Retrieval-Augmented Generation for Multi-Hop Queries, Yixuan Tang and Yi Yang
>
> ### **Does AC-LoRA work without a well-separated embedding space?/ Can AC-LoRA handle overlapping knowledge groups?**
>
>
>
> That's a great point. In AC-LoRA, we utilize an off-the-shelf embedding model to calculate the embedding space and the similarity score, thereby merging the LoRAs.
> For a specific data set where the permission domains overlap, we envision a specialized embedding model being fine-tuned to ensure they are separated.
> However, having overlapping data sets is not detrimental to AC-LoRA because of the following two reasons:
> If one or more overlapping datasets are not included in the user's permitted list, the specific LoRA(s) will not be used to answer the question.
> If one or more are permitted, they will be merged based on the similarity score. Anli_{r1,r2,r3} in FlanV2 are very similar and have a large overlapping embedding space, which we evaluate and cover in this case.
>
> Additionally, we conducted a new experiment to further support our findings. We built a new vectorbase using 20 sets, each combining 2-4 RepliQA topics, resulting in overlapping topics across LoRAs. Since RepliQA requires only one correct LoRA to be retrieved, success rates were high: over 90% for k=1 (retrieving ~3 topics) and up to 99.6% for k=100 (retrieving ~11 topics across ~7 LoRAs).
>
> While these results are expected and only partially meaningful – since no new LoRAs were finetuned and thus end-to-end grades are unavailable due to time constraints – they still should help reinforce the above. A detailed discussion of the required LoRAs training set granularity is beyond the scope of this work.
>
> In conclusion, AC-LoRA is effective for the overlapping permission zones.
>
> *Action point: We will add further discussion to the paper to clarify this.*
>
>
> ### **Is GPU memory a major bottleneck for more than 5/10 LoRAs?**
>
> In our setup, loading the 8B base model with all 17 LoRAs (r=64) requires about 40GB of memory without quantization. Similarly, Flan with 48 LoRAs (r=8) uses only slightly more and still fits on a single 48GB GPU. These requirements depend on factors like the base model size, LoRA configuration (targeted modules, number of layers, rank), and the prompt (and context) length. However, there are parallelization techniques to mitigate this issue, such as pipeline parallelization. This allows AC-LoRA to load more LoRAs.
> Additionally, in a memory-constrained environment, one could keep frequently used LoRAs on the GPU while loading less common ones on demand, trading off a small latency increase.
>
> *Action point: We will add further discussion to the paper to clarify this.*
>
>
> We hope we addressed all your questions and would be happy to continue the discussion to clarify any outstanding concerns.

---

> > ### Comment · Reviewer_WZ6X · 2025-08-01
> >
> > Thank you for the detailed response. Most of my concerns are solved, and I believe the action points listed can increase the clarity of the paper.

---

> ### Author Response · Authors · 2025-08-01
>
> Dear reviewer, thank you for your feedback. We will incorporate the action points into the paper, listed in the rebuttal.

---

### Official Review · Reviewer_jNGe · 2025-07-02

**Clarity:** 4
**Significance:** 3
**Originality:** 2
**Rating:** 4
**Confidence:** 3

**Summary:**

This paper proposes AC-LoRA, a system for developing access control-aware LLM chatbots to prevent leakage of sensitive/confidential data to unauthorized parties. In doing this, the authors attempt to address a gap in existing work where maintaining individual models (or LoRA adapters) for different roles becomes infeasible as the number of permission zones grows, while training a single foundation model compromises data confidentiality. Other approaches such as access-controlled RAG are also computationally expensive and/or detrimental to performance. Instead, the authors propose a method based on fine-tuning LoRAs for individual permission zones. These LoRAs are then mixed together in a permission-ware manner, ensuring that a user does not retrieve information from LoRAs they are not authorized to use; non-permissible LoRAs are then used to provide a hint to the user that they might be able to receive a better answer by applying for relevant permissions. Evaluations are performed using three different datasets: Flanv2 , RepLiQA, and Wikiart. Specifically, the authors show that the proposed LoRA mixing offers competitive performance to perfect selection (using the specific LoRA associated with a given query) and outperforms other baselines in 8 out of 10 evaluated domains.

**Questions:**

- In Table 2, what metric is being reported? Please clarify in the caption.
- Have the authors performed evaluation using queries relevant to multiple domains (i.e., permission zones)?
- How much performance improvement does AC-LoRA offer compared to a simpler strategy of mixing all permissible LoRAs without retrieving relevant ones first?

**Ethical Concerns:**

["NO or VERY MINOR ethics concerns only"]

**Final Justification:**

After reading the authors' response, I believe they have adequately addressed my concerns, and I will maintain my favorable score.

**Limitations:**

The limitations are adequately discussed, with the possible exception of multi-domain queries that require retrieving information from multiple permission zones.

**Paper Formatting Concerns:**

No formatting concerns.

**Quality:**

3

**Strengths And Weaknesses:**

This was an interesting paper addressing an important gap in the usage of LLMs in access controlled environments. The authors propose a plausible training-free solution, and the evaluation results show that AC-Lora performance comparably to perfect selection, while outperforming existing methods in performance and/or latency.

The technical novelty is modest, largely involving a thoughtful combination of existing methods rather than introducing fundamentally new techniques. I would have liked to see additional evaluations, particularly a computational complexity comparison with perfect selection, as well as an ablation study to quantify the performance gains achieved by AC-LoRA compared to simply mixing all permissible LoRAs without the retrieval step. It is also not clear how the framework performs when a query is relevant to multiple domains/LoRAs.

Other more minor comments:
- I suggest the authors revise the Figure captions to make them more self-contained.
- The evaluation using the WikiArt dataset is limited to Figure 7, which does not include any quantitative performance metrics. I recommend either expanding the evaluation for WikiArt or removing it from the evaluation set to make room for other improvements.

---

> ### Author Rebuttal · Authors · 2025-07-29
>
> We thank the reviewer for their time and valuable feedback. In the following, we summarize the main concerns and answer them.
>
> ### **Could you provide the metric in Table 2?**
>
>
> As described on lines 268-269, we use the BLEU score to evaluate the translation, ROUGE for the struct-to-text tasks, and EXACT MATCH for the rest.  We will add this to the caption and revise all figures to make the captions self-contained.
>
>
> ### **Did you evaluate AC-LoRA using queries relevant to multiple domains?**
>
> Yes, we address this with the *CS-Combi* dataset (line 273), where the queries require the knowledge distributed over two LoRAs. This, however, did not test the retrieval capabilities; therefore, we have now generated 100 queries by providing 2-5 RepliQA documents from different topics and asking to build a question that requires all the given documents to answer. We then use these queries to retrieve the relevant LoRAs and verify if they match the topics of the documents.
>
> Below we show some results of the experiments. The number of topics (2-5) is the number of documents used to generate the query. The numbers shown in the first table are the % of the relevant LoRAs retrieved. The second table shows the percentage of correctly retrieved LoRAs. This means that, in general, the percentage increases for more LoRA retrieved (reaching 100% at some point) in the first table, while it decreases in the second table.
>
> | **Number of topics/ % relevant retrieved** | **k** | **# (avg.) Retrieved LoRAs** | **Mean** | **Lower Bound** | **Upper Bound** |
> |:------------------------------------------:|:-----:|------------------------------|:--------:|:---------------:|:---------------:|
> |                      2                     |   50  |             3.73             |   90.3   |       82.6      |       96.1      |
> |                      3                     |   50  |             3.47             |   57.8   |       47.3      |       70.1      |
> |                      4                     |   50  |             4.83             |   59.3   |       51.0      |       66.6      |
> |                      5                     |   50  |             4.77             |   49.0   |       41.2      |       56.7      |
>
>
> | **Number of topics/ % correctly retrieved** | **k** | **# (avg.) Retrieved LoRAs** | **Mean** | **Lower Bound** | **Upper Bound** |
> |:-------------------------------------------:|:-----:|------------------------------|:--------:|:---------------:|:---------------:|
> |                      2                      |   50  |             3.73             |   60.2   |       50.3      |       71.0      |
> |                      3                      |   50  |             3.47             |   56.4   |       46.7      |       67.0      |
> |                      4                      |   50  |             4.83             |   54.2   |       51.0      |       47.1      |
> |                      5                      |   50  |             4.77             |   56.9   |       50.1      |       63.7      |
>
> While we hope that this additional experiment provides further insight, we want to highlight that finding a high-quality dataset with queries that require information from multiple datasets and can be easily separated into different training sets for LoRAs was a major challenge. To this end, we design the dataset *CS-Combi* to evaluate queries that require knowledge from multiple datasets.
> However, a more rigorous evaluation requires manually curating the dataset, which is beyond the scope of this paper.
>
> ### **Is AC-LoRA better than mixing all permissible LoRAs?**
>
> The primary advantage of loading only the relevant LoRA is the reduction of inference latency.
>
> Additionally, we observed an average grade improvement of 0.066 (95% interval: (-0.00748, 0.138)) over all permissible LoRAs in the RepliQA dataset.
> This effect could come from several factors; we suspect it’s primarily due to the relatively high rank used with a relatively small dataset. We expect that the AC-LoRA improvement will increase further as the number of LoRAs increases. Flanv2, however, behaves differently. Since Flanv2 is primarily finetuned to influence output format, mixing all Flan LoRAs together significantly degrades performance. This is evident in the table below, which shows the improvement of AC-LORA over the mixing of all Flan LoRAs.
>
>
> | **FlanV2 / Grade improvement over Mixing All** | **Metric**  | **Mean** | **Lower Bound** | **Upper Bound** |
> |:----------------------------------------:|-------------|:--------:|:---------------:|:---------------:|
> |              struct to text              |   Rouge-1   |   34.64  |      31.95      |      37.19      |
> |              struct to text              |   Rouge-2   |   26.94  |      23.97      |      30.17      |
> |              struct to text              |   Rouge-l   |   30.64  |      28.02      |      33.64      |
> |                commonsense               | Exact Match |   65.5   |       58.5      |       72.5      |
> |                 sentiment                | Exact Match |   90.0   |       85.5      |       94.0      |
> |               reading comp.              | Exact Match |   54.6   |      49.32      |       60.0      |
> |              closed_book QA              | Exact Match |   36.5   |       30.0      |       43.0      |
> |                coreference               | Exact Match |   54.0   |       44.0      |       64.0      |
> |         read.comp.w: commonsense         | Exact Match |   59.0   |      49.97      |       68.0      |
> |                paraphrase                | Exact Match |   63.0   |      56.00      |       69.5      |
> |                    nli                   | Exact Match |   68.2   |      64.44      |      72.32      |
> |                translation               |     Bleu    |    9.7   |       8.22      |      11.43      |
>
>
> ### **Can you provide quantitative performance metrics for WikiArts?**
>
> We do not provide empirical results for WikiArts, as it serves primarily as a proof-of-concept for AC-LoRA extending beyond text. The limitations of multi-modal evaluation are addressed in Appendix C3 (line 721). While we could evaluate the retrieval, this does not add any insight as it is already covered by the other text-based datasets. Additionally, we would need to generate a suitable test set, as none currently exist. Regarding image quality, there is no practical way to measure it, since the base model has likely been trained on WikiArts or similar images, and a human study is beyond the scope of this paper.
>
> *Action point: We can highlight this in the paper more, or if necessary, remove it from the main body of the paper.*
>
>
> We hope we addressed all your questions and would be happy to continue the discussion to clarify any outstanding concerns.

---

### Official Review · Reviewer_VTRK · 2025-07-04

**Clarity:** 3
**Significance:** 3
**Originality:** 2
**Rating:** 3
**Confidence:** 4

**Summary:**

This paper introduces a new method called AC-LORA, which tackles the problem of letting a single large model serve many users who each see only the data they are entitled to. Instead of fine-tuning one big model on all documents, or leaving those documents into the prompt, it fine-tunes a lightweight LoRA adapter for every permission segment, stores each adapter’s embedding in a vector index, and at inference time retrieves only the adapters that are semantically relevant to the query and  allowed for the current user. These adapters are loaded on-the-fly and blended by cosine-similarity weighting, so no extra routing network or retraining is needed when new data arrive. Experiments on text (RepLiQA, Flan-V2) and image generation (WikiArt styles) show that AC-LORA matches or beats task-specific LoRA baselines while slashing context length, latency, and leakage risk. The system therefore offers a practical, nearly “zero-additional-training” way to combine access control, retrieval, and knowledge fusion for multi-modal LLM applications, with remaining challenges around hint privacy and adapter-swap overhead in very large deployments.

**Questions:**

1. Please give analytical or empirical evidence that simple cosine-weighted blending is robust when two adapters make conflicting updates to the same weight tensor, visualise conflict rates or run controlled ablations.

2. Please attach 95 % confidence intervals or formal significance tests (e.g., paired t-tests or bootstrap) for the main RepLiQA, Flan-V2, and WikiArt results and state the random seeds used.

**Ethical Concerns:**

["NO or VERY MINOR ethics concerns only"]

**Final Justification:**

I appreciate the response and new experiments from authors, which address part of my concerns. However, it is still not convinced by the explanation of weight tensor conflict, Hint mechanism, and inference latency with more LoRAs. I will keep the score.

**Limitations:**

The paper briefly admits that the Hint mechanism could leak sensitive information but never formalizes any threat model or measures what is actually exposed, and it overlooks broader risks such as adapter-fusion bypassing content filters, the environmental cost of large-scale hot-swapping, and governance concerns around data revocation or labor displacement. A concise yet concrete treatment of these issues like quantitative leakage metrics, energy estimates, and responsible-deployment guidelines would make the societal-impact discussion far more convincing.

**Quality:**

3

**Strengths And Weaknesses:**

Strengths:
1. This paper combines LoRA tuning, vector search, and cosine mixing into a single, low-overhead framework, which is a novel integration. 2. This mode works across modalities, which is shown to run on both text (LLAMA-2/3) and image models (Stable-Diffusion, Qwen-VL).
3. It provides fine-grained access control and hot-swappable adapters, matching real-world deployment needs.
4. Leakage-extraction attacks confirm that per-adapter isolation greatly reduces verbatim data leaks, demonstrating the security.

Weaknesses:
1. No confidence intervals or significance tests, so robustness of the gains remains uncertain.
2. Results stop at ten adapters; latency and memory for hundreds are not reported.
3. This paper offers little analysis of when linear LoRA mixing could fail due to conflicting updates.

---

> ### Author Rebuttal · Authors · 2025-07-29
>
> We thank the reviewer for their time and valuable feedback. In the following, we summarize the main concerns and answer them.
>
> ### **Conflicting updates to the same weight tensor**
>
> AC-LoRA addresses the issue of conflicting updates to the same weight tensor, as highlighted by the reviewer. Conflicting weight merging is a known challenge in existing works that combine multiple LoRAs by merging their weights, resulting in accuracy loss while merging more than two LoRAs [1].
>
> However, AC-LoRA *does not* rely on merging weights. Instead, it mixes the outputs of each layer based on cosine similarity between the user query and permitted LoRA datasets. As a result, even if the retrieved LoRAs make a conflicting update to the same weight tensor,  AC-LoRA remains unaffected by these conflicts.
> Intuitively, the final output can be seen as the output of a higher-ranked LoRA, which is composed of the retrieved LoRAs.
>
> *Action point: We will clarify this in Section 3.*
>
> [1] LoraRetriever: Input-Aware LoRA Retrieval and Composition for Mixed Tasks in the Wild, Zhao et al.
>
> ### **95% confidence intervals for the main experiments, and the random seed**
>
> The (bootstrapped) 95% confidence intervals of all our major experiments in the paper are as follows.
>
>
> **RepLiQA**
>
> |   **RepLiQA Topics / Grades**   | **Mean** | **Lower Bound** | **Upper Bound** |
> |:-------------------------------:|:--------:|:---------------:|:---------------:|
> |               ALL               |   2.214  |      2.161      |      2.266      |
> |       Local Environmental Issues  |   2.352  |      2.1118     |      2.592      |
> |    Local Health and Wellness    |   2.307  |      2.104      |      2.509      |
> |        Cybersecurity News       |   2.151  |      1.940      |      2.378      |
> |   Local Sports and Activities   |   2.179  |      1.964      |      2.381      |
> |      Local Arts and Culture     |   2.035  |      1.858      |      2.225      |
> |       Neighborhood Stories      |   1.979  |      1.786      |      2.172      |
> |         Company Policies        |   2.882  |      2.611      |      3.152      |
> |            Local News           |   2.262  |      2.051      |      2.468      |
> |     Local Education Systems     |   2.140  |      1.929      |      2.359      |
> |    Regional Folklore and Myth   |   2.222  |      2.023      |      2.452      |
> |         Incident Report         |   2.193  |      1.993      |      2.393      |
> |           News Stories          |   2.038  |      1.839      |      2.251      |
> | Local Technology and Innovation |   1.920  |      1.712      |      2.122      |
> |   Regional Cuisine and Recipes  |   2.380  |      2.156      |      2.604      |
> |  Local Politics and Governance  |   2.127  |      1.906      |      2.338      |
> |  Local Economy and Market       |   2.434  |      2.204      |      2.663      |
> |   Small and Medium Enterprises  |   2.297  |      2.081      |      2.506      |
>
> For k = 2: (others will be added to the paper)
>
> | **RepLiQA Topics / Retrival %** | **Mean** | **Lower Bound** | **Upper Bound** |
> |:-------------------------------:|:--------:|:---------------:|:---------------:|
> |               ALL               |   89.4   |       88.5      |       90.3      |
> |    Local Environmental Issues   |   89.6   |       85.2      |       93.5      |
> |    Local Health and Wellness    |   92.2   |       88.8      |       95.7      |
> |        Cybersecurity News       |   91.6   |       88.0      |       95.3      |
> |   Local Sports and Activities   |   93.0   |       89.5      |       96.0      |
> |      Local Arts and Culture     |   93.0   |       89.7      |       96.2      |
> |       Neighborhood Stories      |   79.0   |       74.2      |       83.8      |
> |         Company Policies        |   91.9   |       86.8      |       96.3      |
> |            Local News           |   78.3   |       72.9      |       83.7      |
> |     Local Education Systems     |   89.2   |       85.0      |       93.4      |
> |    Regional Folklore and Myth   |   95.6   |       92.6      |       98.0      |
> |         Incident Report         |   91.2   |       87.7      |       94.7      |
> |           News Stories          |   90.8   |       86.9      |       94.6      |
> | Local Technology and Innovation |   85.5   |       80.5      |       90.0      |
> |   Regional Cuisine and Recipes  |   95.6   |       92.9      |       97.8      |
> |  Local Politics and Governance  |   89.2   |       84.8      |       93.1      |
> |     Local Economy and Market    |   81.7   |       76.6      |       86.9      |
> |   Small and Medium Enterprises  |   93.0   |       89.5      |       96.5      |
>
>
> **FLAN**
>
> |        **FlanV2**        |  **Metric** | **Mean** | **Lower Bound** | **Upper Bound** |
> |:------------------------:|:-----------:|:--------:|:---------------:|:---------------:|
> |      struct to text      |   rouge-1   |   61.4   |       58.2      |      64.76      |
> |      struct to text      |   rouge-2   |   37.14  |      33.83      |      40.27      |
> |      struct to text      |   rouge-l   |   54.23  |      51.17      |      57.31      |
> |        commonsense       | Exact Match |   65.5   |      58.48      |      72.01      |
> |         sentiment        | Exact Match |   90.0   |       86.0      |       94.0      |
> |       reading comp.      | Exact Match |   54.6   |       49.0      |      60.66      |
> |      closed_book QA      | Exact Match |   36.5   |       30.5      |      43.01      |
> |        coreference       | Exact Match |   54.0   |       44.0      |       64.0      |
> | read.comp.w: commonsense | Exact Match |   59.0   |       49.0      |       69.0      |
> |        paraphrase        | Exact Match |   63.0   |      56.49      |       69.5      |
> |            nli           | Exact Match |   68.2   |      64.24      |      72.52      |
> |        translation       |     Bleu    |   13.4   |      11.37      |      15.45      |
>
> For k=2 (others will be added to the paper)
>
> |  **FlanV2 / Retrival %** | **Mean** | **Lower Bound** | **Upper Bound** |
> |:------------------------:|:--------:|:---------------:|:---------------:|
> |      struct to text      |   90.0   |       85.5      |       94.0      |
> |        commonsense       |   65.0   |       58.0      |       71.0      |
> |         sentiment        |   95.5   |       92.5      |       98.0      |
> |       reading comp.      |   58.0   |       52.3      |       63.6      |
> |      closed_book QA      |   21.5   |       16.0      |       27.0      |
> |        coreference       |   81.0   |       73.0      |       88.0      |
> | read.comp.w: commonsense |   77.0   |       68.0      |       85.0      |
> |        paraphrase        |   95.5   |       92.5      |       98.0      |
> |            nli           |   56.1   |       51.5      |       60.6      |
> |        translation       |   85.0   |       81.2      |       88.5      |
>
>
> We do not provide empirical results for WikiArts, as it serves primarily as a proof-of-concept for AC-LoRA extending beyond text. The limitations of multi-modal evaluation are addressed in Appendix C3 (line 721). Regarding image quality, there is no practical way to measure it, since the base model has likely been trained on WikiArts or similar images, and a human study is beyond the scope of this paper.
>
> The *random seed used was 42*.
>
> *Action point: We will update the paper accordingly.*
>
> ### *Threat model and leakage evaluation for the hint mechanism**
>
> As the reviewer pointed out, we did not thoroughly discuss the *hint* feature. We believe that, besides providing the correct answer, a corporate AI agent must also assist the user in applying for document permission in case the user does not have access to the most relevant document. The maintainer/administrator can decide if the user will be permitted to access it. The hint provides a link to the document permission, e.g., access permission to a Git repository. However, a curious/malicious user can gain knowledge of the possible *existence* of the information. Disabling the hint for sensitive LoRAs (e.g., specification of an upcoming product) will prevent AC-LoRA from mentioning the existence of a specific dataset to not-permitted users. However, measuring the exact potential for leakage requires creating a particular dataset and benchmark to reflect such a scenario.
>
> *Action point: We will add a discussion of this topic both in the threat model and AC-LoRA design.*
>
> ### **The mixing of Adapters to bypass content filters**
>
> This is an interesting question, and the response largely depends on the type of content filter. Assuming that the content filtering is done by aligning the base model in a specific way [1], we will outline three different scenarios in which they could be affected by AC-LoRA:
>
> - *No LoRAs are retrieved*: In case the user query does not match any LoRA, no LoRA will be retrieved, and the base model will answer the query. In that case, the original alignment of the model remains unchanged.
> - *LoRA is retrieved*: If LoRA(s) are retrieved, it's challenging to ensure model alignment without considering this during fine-tuning.
> - *Specific LoRAs for bypassing filters*: A set of LoRAs could be specifically fine-tuned to bypass the content filter (e.g., LoRAs used by law enforcement), making the bypass of the filter intentional.
>
> However, we agree that the broader impact of such a system is complex, requires further analysis, and is beyond the scope of this paper.
>
> *Action point: We will add discussions to address this.*
>
>
> [1] Llama Guard: LLM-based Input-Output Safeguard for Human-AI Conversations, Hakan Inan et al.
>
> ### **>100s LoRAs**
>
> Inference latency and VRAM requirement increase with the number of loaded LoRA. Due to the absence of a dataset and GPU resources, we did not experiment with such a scale (please see *reviewer WZ6X* answer 3 for more information).
>
> We hope we addressed all your questions and would be happy to continue the discussion to clarify any outstanding concerns.

---

> ### Author Response · Authors · 2025-08-05
>
> Dear reviewer, thank you for providing the valuable comments. In our rebuttal, we address the questions and concerns that you have raised. Let us know if the rebuttal adequately answers your questions. We are happy to have further discussions and clarify any remaining issues.

---

### Note · Authors · 2025-08-13

Dear AC and reviewers,

We thank the reviewers for taking the time to review the paper and for providing valuable feedback. The questions raised by the reviewers led to additional experiments, resulting in additional insights that strengthen the paper and highlight its contributions. All the questions and concerns raised by the reviewers have been answered in the rebuttal with further experiments and reasoning, and we are happy to inform you that there are no new concerns raised by the reviewers.

In summary, AC-LoRA provides a valuable step forward in understanding how to enforce strong information isolation in LLMs using LoRAs fine-tuned with disjoint datasets. While the concept of LoRA merging is well known in the community,  AC-LoRA’s approach of using *training-free* LoRA mixing for *access control* is unique for *multi-modal models* and has not been covered in the literature. AC-LoRA also establishes that the knowledge injection through LoRA is feasible and critical for imposing access control primitives in state-of-the-art LLMs. Additionally, AC-LoRA provides a complete *end-to-end proof-of-concept* with open-source models and datasets, demonstrating the practicality of such a system. Therefore, AC-LoRA opens a new research direction to investigate how such information isolation and access control mechanisms for LLM can be extended to other scenarios, specifically for increasing model safety and reducing harm.

---

### Decision · Program_Chairs · 2025-09-17

**Decision:**

Accept (poster)

**Comment:**

- This paper proposes an access-control variant of LoRA where different LoRAs are trained on different permissioned datasets and used by matching the user's prompt and permission.
- Overall, it is a well-motivated and professionally written paper with good design and sufficient experiments.
- Reviews are mostly positive, with only one borderline reject (that does not include strong arguments to reject the paper, and their concerns are mostly addressed by the rebuttal). I would ask the authors to include the additional experiments and analysis in the next paper version.